# Recurrent evolution of adhesive defence systems in amphibians by parallel shifts in gene expression

Shabnam Zaman [1], Birgit Lengerer[2,3], Joris Van Lindt[4], Indra Saenen[1], Giorgio Russo[4], Laura Bossaer [1], Sebastien Carpentier[5], Peter Tompa [4,6], Patrick Flammang [2] & Kim Roelants [1] ✉

Natural selection can drive organisms to strikingly similar adaptive solutions, but the underlying molecular mechanisms often remain unknown. Several amphibians have independently evolved highly adhesive skin secretions (glues) that support a highly effective antipredator defence mechanism. Here we demonstrate that the glue of the Madagascan tomato frog, *Dyscophus guineti*, relies on two interacting proteins: a highly derived member of a widespread glycoprotein family and a galectin. Identification of homologous proteins in other amphibians reveals that these proteins attained a function in skin long before glues evolved. Yet, major elevations in their expression, besides structural changes in the glycoprotein (increasing its structural disorder and glycosylation), caused the independent rise of glues in at least two frog lineages. Besides providing a model for the chemical functioning of animal adhesive secretions, our findings highlight how recruiting ancient molecular templates may facilitate the recurrent evolution of functional innovations.

Biological adhesives are a widespread adaptation in the Animal Kingdom that have arisen multiple times in many distantly related taxa[1]. They serve a variety of functions essential for survival, from long-term substrate attachment (e.g., adhesive proteins of mussel byssus, barnacle and tubeworm cement[2–4]) to locomotion (e.g., echinoderm tube feet secretion[5]), development (e.g., silk moth cocoons[6]) and prey capture (e.g., spider silk threads[7], velvet worm slime[8]). Some of these adhesive secretions have been intensively studied and their components, as well as the mechanisms through which they interact, are now well-characterised (e.g., DOPA (3,4-dihydroxyphenylalanine)-containing mussel foot proteins[9]; spider silk spidroins[10]). However, as most of this research has had a focus on medical and biotechnological

applications, the evolution of adhesive secretions as a recurrent innovation throughout the animal tree remains poorly understood.

It is widely known that the skin of many amphibians supports a defence function against aggressors[11]. Skin-secreted poisons constitute the most pervasive antipredator adaptation in this vertebrate clade and typically contain cocktails of diverse toxins. However, some amphibians produce skin secretions with a different, highly effective defence mechanism. When attacked by a predator, a small number of frog and salamander species discharge a viscous fluid from their skin that quickly solidifies into a sticky mass[12–16] (hereafter called a glue). The adhesive and frictional forces exerted by such glues dramatically increase the energetic cost of prey handling. For a predator with

[1]Ecology, Evolution & Genetics Research Group (bDIV), Biology Department, Vrije Universiteit Brussel, Pleinlaan 2, 1050 Brussels, Belgium. [2]Biology of Marine Organisms and Biomimetics Unit, Research Institute for Biosciences, University of Mons, Place du Parc 23, 7000 Mons, Belgium. [3]Evolutionary and Developmental Biology, Department of Zoology, University of Innsbruck, Technikerstr. 25, 6020 Innsbruck, Austria. [4]Center for Structural Biology, VIB-VUB and Structural Biology Brussels, Vrije Universiteit Brussel, Pleinlaan 2, 1050 Brussels, Belgium. [5]Proteomics Core - SyBioMa, Katholieke Universiteit Leuven, Herestraat 49 – 03.313, 3000 Leuven, Belgium. [6]Institute of Molecular Life Sciences, HUN-REN Research Centre for Natural Sciences, Magyar Tudósok Körútja 2, 1117 Budapest, Hungary. ✉e-mail: Kim.Roelants@vub.be

limited dexterity (such as a snake), ingestion likely becomes an insurmountable task, forcing it to release the amphibian and thus enabling the latter's survival[17]. Although the glues of some amphibians are also poisonous[18], others may have lost toxicity[19], suggesting that stickiness is a suitable alternative chemical defence weapon.

Skin-secreted glue is shared by genera in phylogenetically distant amphibian lineages, including brevicipitid, hylid, alsodid, microhylid and myobatrachid frogs[12–15] and plethodontid and ambystomatid salamanders[16,20]. As close relatives of these glue-secreting taxa often produce nonadhesive (and often toxic) skin secretions, glue represents an example of parallel functional innovation: one that changes the phenotype at a molecular scale but manifests its effect - antipredator defence - at the macroscopic level. Although the adhesive strengths of a few amphibian secretions have been examined[12], most aspects of the glues they produce are still unknown, raising questions about the genetic mechanisms underpinning its recurrent evolution from a nonadhesive poison. A proteinaceous nature of the glues of the frog species *Notaden bennetti* (Myobatrachidae)[13] and *Eupsophus vertebralis* (Alsodidae)[15] has been demonstrated and while a protein called Nb1 has been reported for the former species, it was never characterised in detail. Studies on the adhesive secretion of another glue-secreting amphibian, the Madagascan tomato frog *Dyscophus guineti* (Microhylidae), identified one of its constituents as a serine protease inhibitor[21,22]. Its role in the glue, however, remains speculative[22].

Here, we conducted transcriptomic and proteomic analyses of the skin and adhesive secretions of *D. guineti* to identify proteins central to its stickiness. Characterisation of these proteins allows us to propose a general model for their interactions that may also be relevant to other biological adhesive systems. Subsequent comparative analyses with homologous proteins of other amphibians reveal the structural changes required to transform a nonadhesive toxic defence secretion into an adhesive one. Our findings show that, once a suitable glue protein had evolved, elevated gene expression proved instrumental in the recurrent emergence of adhesive secretions in multiple lineages.

## Results

### The adhesive skin secretion of *D. guineti* has a proteinaceous basis

Physical handling of *D. guineti* frogs initially results in the production of a slippery exudate across the body and limbs. This substance, henceforth called mucus, is not adhesive and resembles the skin secretions of many other amphibians. Prolonged stimulation causes the animal to inflate and produce a viscous white fluid from its back (but not its limbs or ventral region, which continue to secrete the slippery mucus). This material, hereafter referred to as glue, becomes adhesive upon contact with any foreign object.

Mechanical pull tests were carried out using smooth-surfaced hard plastic blocks (LEGO® bricks; Supplementary Fig. 1) which were joined together with freshly collected *D. guineti* glue at a pressure of 40 kPa (i.e., within the range reported for the bite of an average-sized snake expected to prey on these frogs[23]). The strength of this bond between two bricks, or tensile strength, was calculated as the pull force at failure (when the joined bricks break apart) divided by the area of the bonded surface. The mean tensile strength was $33.62 \pm 13.36$ kPa ($n = 7$) after 10 minutes of curing, but increased to $65.51 \pm 37.24$ kPa ($n = 15$) after 60 minutes of curing (Fig. 1a). These strengths are comparable to those reported for glues of other frog species[12,13], fitting the expected pattern of a recurrently evolved antipredator system. In contrast, the dorsal skin secretion of *Bombina orientalis*, known to be poisonous but nonadhesive, consistently failed to glue the blocks together for either length of time (i.e., tensile strength of 0 kPa). Joining two bricks with *D. guineti* glue at a reduced pressure of 4 kPa resulted in a lower tensile strength after 60 minutes of curing ($26.47 \pm 13.65$ kPa; $n = 7$). This finding indicates that the tensile strength of *D. guineti* glue is pressure sensitive.

To investigate this further, we visualised the ultrastructure of *D. guineti* glue using scanning electron microscopy (SEM) after applying low vs. moderate manual pressure on the glue (i.e., dipping vs. pressing a microscope coverslip in freshly secreted glue). The resulting images reveal the abundance of secretory granules of approximately 1 µm in diameter, suspended in a mesh-like glue matrix (Fig. 1b). Increased pressure resulted in the release of much more of the glue from the granules. This finding suggests that pressure enhances the release of glue material from these granules after their intact secretion from the skin glands. Visual inspection of the separated bricks after 10 minutes of curing shows comparable glue traces on both bricks (Fig. 1c), indicating that the break was caused by failure of cohesive bonding within the glue mass. However, after 60 minutes of curing, the glue in the majority of replicates remained mostly attached to a single brick, while leaving large patches of clean surface on the other brick (Fig. 1c). This pattern suggests that breaking was now predominantly caused by failure of adhesive bonding. Together, these observations suggest that cohesive strength increases at a slower rate than adhesive strength during the curing of *D. guineti* glue, but eventually reaches higher levels.

Gel electrophoresis confirms the presence of multiple proteins in both glue and mucus, ranging in size from about 10 kDa to over 400 kDa in both secretions (Supplementary Fig. 2a). The presence of similar gel bands suggests that some proteins may be shared between both secretion types, but their concentration in mucus appears to much lower than in glue. Treatment of *D. guineti* glue with two serine proteases, proteinase K and trypsin, lead to complete loss of tensile strength (Fig. 1a), indicating that its adhesive and/or cohesive properties are regulated by proteins. In addition, lectin labelling of the glue revealed the presence of glycoproteins in the secretion, and oligosaccharide moieties (glycans) including galactose, fucose, *N*-acetylgalactosamine (GalNAc) and *N*-acetylglucosamine (GlcNAc) residues were recognised (Supplementary Fig. 3). The proteinaceous nature of the glue implies that it is gene-encoded, allowing investigation of its origin and adaptive evolution at the genetic level.

### The key component of *D. guineti* glue is an intrinsically disordered glycoprotein

To identify proteins underlying the adhesive property of *D. guineti* skin secretion, we constructed an RNA-seq transcriptome library of its dorsal skin. Besides providing an overview of expressed genes, this library served as a database to analyse proteomic (mass spectrometry; MS) data obtained from glue samples (produced by dorsal skin) and compare them to those of mucus samples (produced by limb skin). Using this approach, we found transcriptional confirmation of the previously described *D. guineti* serine protease inhibitor[21,22]. However, proteomic analyses did not substantiate the presence of the serine protease inhibitor in either the glue or mucus, perhaps because of our choice of proteomic methods (see Methods) or due to the previously reported low levels of this protein in the secreted material[22]. Similarly, we found no evidence for the secretion of any bioactive peptides shared by a wide range of other frog families (including neuropeptides, hormone analogs or cytolytic peptides[11]). Although the presence of nonproteinaceous toxins (e.g., alkaloids or steroids) cannot be excluded, the absence of peptides seems to suggest that the skin secretion of *D. guineti* lost its toxicity either before, during, or after acquiring its adhesive property.

Instead, another protein emerged as an abundant component of the glue and thus as a prime candidate for a glue-associated function. Since this protein contains multiple copies of IgGFc binding domain (InterPro: IPR035234), a domain located at the N-terminus of extracellular proteins in a wide range of metazoans[24], we name it PRIT-Dg (**P**rotein with **R**epeated **I**gGFcBD in **T**andem - *Dyscophus guineti*). We use the acronym IgGFcBD strictly to refer to this domain, and not the IgGFc binding protein (IgGFcBP; Uniprot: Q9Y6R7) to which it lends its

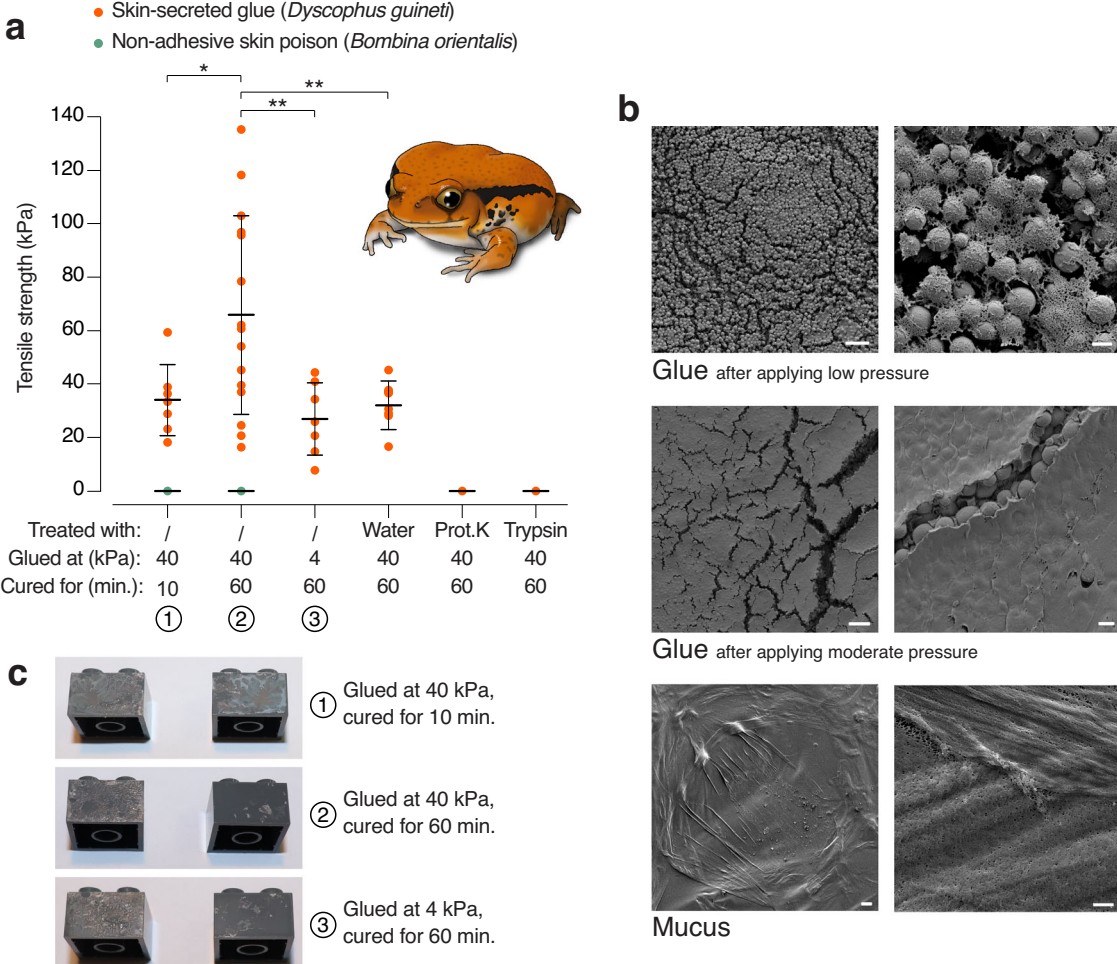

**Fig. 1 | The glue of *Dyscophus guineti* relies on interacting proteins. a.** While a reference skin secretion (*B. orientalis* poison; green data points) shows no measurable tensile strength, *D. guineti* glue (red data points) attains a tensile strength that depends on curing time ($n = 7$ vs. $n = 15$; linear mixed model, $z = -2.324$, $p = 0.0201$) and applied pressure ($n = 7$ vs. $n = 15$; linear mixed model, $z = 2.911$, $p = 0.0036$). Incubation of glue in water reduced tensile strength ($n = 8$ vs. $n = 15$; linear mixed model, $z = -2.58$, $p = 0.0099$) but similar incubation with proteinase K or trypsin (both at 2 mg/ml) totally eliminated it, implying a proteinaceous basis. Probabilities are from two-sided $z$-tests and are not corrected for multiple comparisons. All measured data points are shown; bars present mean ± SD; asterisks denote significant differences (* = $p < 0.05$; ** = $p < 0.005$). **b** Scanning electron microscopy (SEM) images of secreted *D. guineti* glue after applying low pressure or moderate pressure and of secreted mucus. The mesh-like matrix, in which spherical granules are suspended, increases in abundance with increased pressure. Scale bars represent 10 μm (left) or 1 μm (right). **c** Photographs of LEGO® bricks after being separated by pull tests indicate failure of cohesive bonding after 10 min of curing and of adhesive bonding after 60 min.

name. Homology searches against public protein databases using PRIT-Dg as a query yielded mostly functionally uncharacterised hits in other vertebrates. One exception involves a hit described as a peptidylaminoacyl-L/D-isomerase in the poison of *Bombina* frogs[25] (Uniprot: Q58PK6; GenBank: AAX55674; e-value = $2e^{-101}$), implying a different function in nonadhesive skin secretions.

Transcriptome assemblies using different approaches unanimously identify the gene encoding PRIT-Dg as one of the highest expressed in *D. guineti* dorsal skin. Consistently, normalised MS data indicate that PRIT-Dg is the most prevalent protein in glue, with levels ranging between nine to 37 times higher than in mucus (Supplementary Fig. 4a). Immunological analyses with antibodies specifically raised against PRIT-Dg confirmed its abundance in glue and, conversely, its near-absence in mucus (Fig. 2a). In addition, applying manual pressure on the glue prior to immunolabelling resulted in bright confluent patches of fluorescence, suggesting that more of the protein was exposed, consistent with its increased release from secretory granules under pressure. Immunohistochemistry of dorsal skin sections using the same antibodies further demonstrated that, prior to secretion, PRIT-Dg is localised in dorsal granular glands

(typically producing bioactive proteins) but not in mucous glands (typically producing mucus-related proteins; Supplementary Fig. 5). Together, these results support a glue-specific role for PRIT-Dg in *D. guineti* adhesive secretion.

With the primary glue protein identified, we proceeded to characterise the sequence of PRIT-Dg and determine how it may be involved in the secretion's adhesive functioning. Concordant with our lectin stains (Supplementary Fig. 3), immunoblotting of glue before and after treatment with deglycosylation enzymes confirms that PRIT-Dg is indeed a glycoprotein (Supplementary Fig. 6). In an attempt to obtain the full-length sequence of PRIT-Dg, we used the PRIT-encoding transcripts in our RNA-seq library to design specific primers for the amplification of cDNA reverse-transcribed from *D. guineti* dorsal skin tissue (see Methods). This approach resulted in multiple amplicons ranging from 1.9 kb to upwards of 20 kb. Since nine of these amplicons share a near-identical 895 bp-long segment at their 3' ends, they may represent alternative splice variants encoding PRIT-Dg isoforms of different lengths. Due to the prevalence of repeats (see further), the full sequence of these transcripts could not be recovered even with the use of different approaches (RNA-seq, RACE-PCR, long-range PCR,

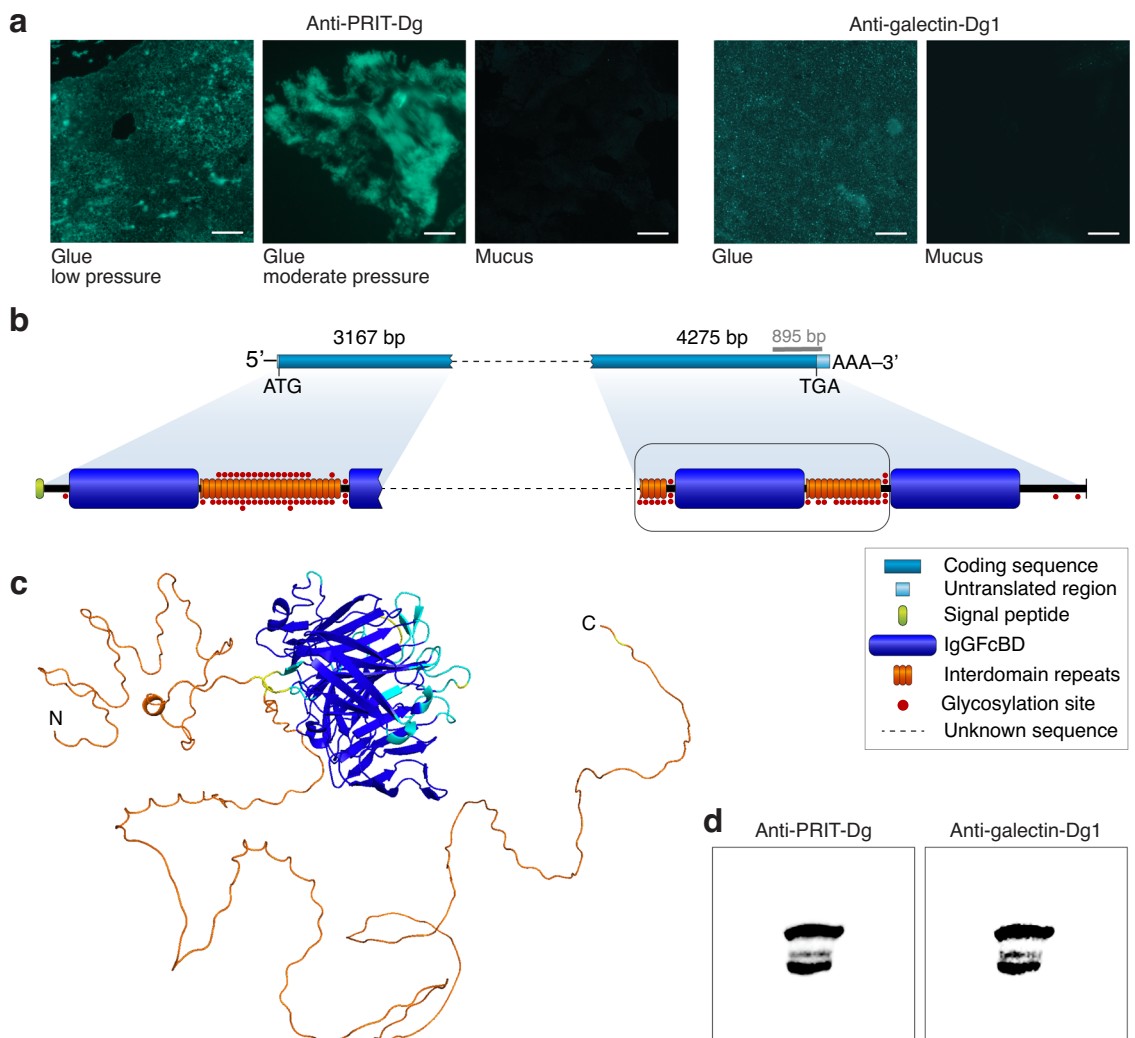

**Fig. 2 | Characterisation of the newly identified glue protein PRIT-Dg and interaction with the co-secreted galectin-Dg1. a** Immunoreactivity assays with antibodies specific to PRIT-Dg and galectin-Dg1 confirm their presence in *D. guineti* glue but not in mucus. Scale bars represent 50 µm. **b** Partial sequencing of a 10-kb transcript reveals a secretory protein with an N-terminal signal peptide (green), followed by multiple IgGFc binding domains (IgGFcBD; blue) that are interspersed with intrinsically disordered repeats (orange). **c** AlphaFold-based structural model of a single IgGFcBD (blue) of PRIT-Dg flanked on both sides (N- and C-termini; orange) by long structurally heterogeneous regions of intrinsically disordered repeats. The prediction is coloured using the per-residue confidence metric pLDDT, with blue denoting regions modelled to high accuracy and orange denoting regions of low prediction confidence, often corresponding to intrinsically disordered segments. **d** Under native conditions, PRIT-Dg interacts with co-secreted galectin-Dg1, as indicated by sequential western blotting with antibodies specific for *(1)* the interdomain repeats of PRIT-Dg and *(2)* galectin-Dg1. Bands depict the same position on a single membrane which was first blotted with anti-PRIT-Dg antibodies, then stripped, and finally blotted with anti-galectin-Dg1 antibodies. Different band sizes may represent alternative PRIT-Dg isoforms, all of which interact with galectin-Dg1. Since protein behaviour in terms of migration in a native PAGE environment is unknown, molecular weights of interacting proteins cannot be reliably estimated, and a reference protein ladder has thus not been provided.

primer walking). Nevertheless, for the ≈ 10-kb transcript, we sequenced the first 3,167 bp of its 5'-side and 4,275 bp of its 3'-side, together covering approximately 70% of its coding sequence (Fig. 2b). We therefore roughly estimate that the full protein encoded by this transcript variant is approximately 3250 amino acids long, corresponding to a molecular weight of ≈ 358 kDa (Supplementary Fig. 2b).

The amino-acid sequence of PRIT-Dg harbours a large proportion of glycine (13.7%), proline (11.9%), isoleucine (10.1%) and lysine (9.0%) residues and a low cysteine content (0.5%). Based on our assembled sequences, we predict that PRIT-Dg isoforms feature between one and at least four IgGFcBD which are approximately 390 amino acids in length and separated by an intervening segment of 12 to 24 imperfect tandem repeats. Each repeat unit spans 18 amino acids and includes up to three serine or threonine residues predicted to serve as GalNAc-type *O*-linked glycosylation sites, and up to four tandem proline-glycine

(PG) dipeptide motifs. Notably, all repeats are predicted to be highly intrinsically disordered (100% of residues): a characteristic of protein regions with sequences that preclude stable folding, rendering them structurally heterogenous[26]. These predictions of structural disorder are reinforced by structural modelling of a single PRIT-Dg module using AlphaFold[27], which reconstructs a discrete, structured IgGFcBD domain flanked by two segments of low prediction confidence, indicative of structural disorder (Fig. 2b). These sequence-based inferences thus complete the picture of a modular, highly disordered and heavily glycosylated protein.

Based on the abundance of glycoproteins highlighted by our lectin labelling (Supplementary Fig. 3), we predicted that carbohydrate-binding proteins (lectins) may also serve a role in the *D. guineti* adhesive secretion, similar to the biological adhesives of other organisms (e.g., sea stars[28]). The dorsal skin transcriptome indeed

contains transcripts of multiple galectins (InterPro: IPR044156): ubiquitous proteins that, through their ability to bind glycans, facilitate various extracellular cross-linking functions. Two of these transcripts, galectin-Dg1 and galectin-Dg2, rank alongside PRIT-Dg as the highest expressed in the library. The high expression of galectin-Dg1 is corroborated by MS data, which shows that the corresponding protein is produced at much higher (four- to 126-fold) levels in glue than in mucus (Supplementary Fig. 4b). Similarly, labelling with antibodies specifically directed against galectin-Dg1 confirmed protein abundance in the adhesive secretion (Fig. 2a) and dorsal granular glands (Supplementary Fig. 7). The sequence of galectin-Dg1, composed of 145 amino acids, contains a single carbohydrate recognition domain (CRD), as defined by a signature amino-acid motif conserved among vertebrate galectins[29]. This single CRD defines galectin-Dg1 as a mono-CRD (prototypical) galectin which, given the group's tendency to form noncovalent homodimers[30], may contribute to the glue's functioning as a cross-linking protein.

The fundamental properties of PRIT-Dg and galectins (i.e., a glycoprotein and a class of carbohydrate-binding proteins, respectively) allude to a possible protein-protein interaction that may reinforce the glue's inherent cohesive strength. Compellingly, the PRIT-galectin interaction was recovered intact when antibodies specific to PRIT-Dg and galectin-Dg1 were used under nondenaturing conditions (native PAGE), signifying that they do in fact interact within the glue milieu (Fig. 2c). We further confirmed this association using co-immunoprecipitation experiments, whereby antibodies targeting one protein systematically revealed the presence of the other as a binding partner and vice versa (Supplementary Fig. 8). These results together provide the essential elements to conceptualise a model for the functioning of an amphibian glue.

## Elevated gene expression underlays the parallel origins of defence glues

The identification of glue proteins allows us to examine the genetic changes that gave rise to the evolution of a bioadhesive. To do so, we complemented comparative transcriptome analyses of a diverse set of amphibians with genomic and phylogenetic analyses including a broader range of taxa. These analyses reveal that several structural features of PRIT-Dg, as well as its expression in skin, evolved in an ancestral gene before the actual origin of a defence glue.

Transcriptome analyses indicate that skin expression of an IgGFcBD-containing protein is not unique to *D. guineti* and dates back to an early amphibian ancestor. We find transcripts encoding IgGFcBD-containing proteins in the skin libraries of six diverse amphibians that produce nonadhesive poisons (see further), as well as in the skin library of *Breviceps mossambicus*, a representative of a frog lineage whose secretion evolved into a glue independently from that of *D. guineti*. Genomic screening of a wide range of taxa identified genes encoding IgGFcBD-containing proteins in all major vertebrate lineages (Fig. 3a) but also in echinoderms, hemichordates and tunicates[24]. In

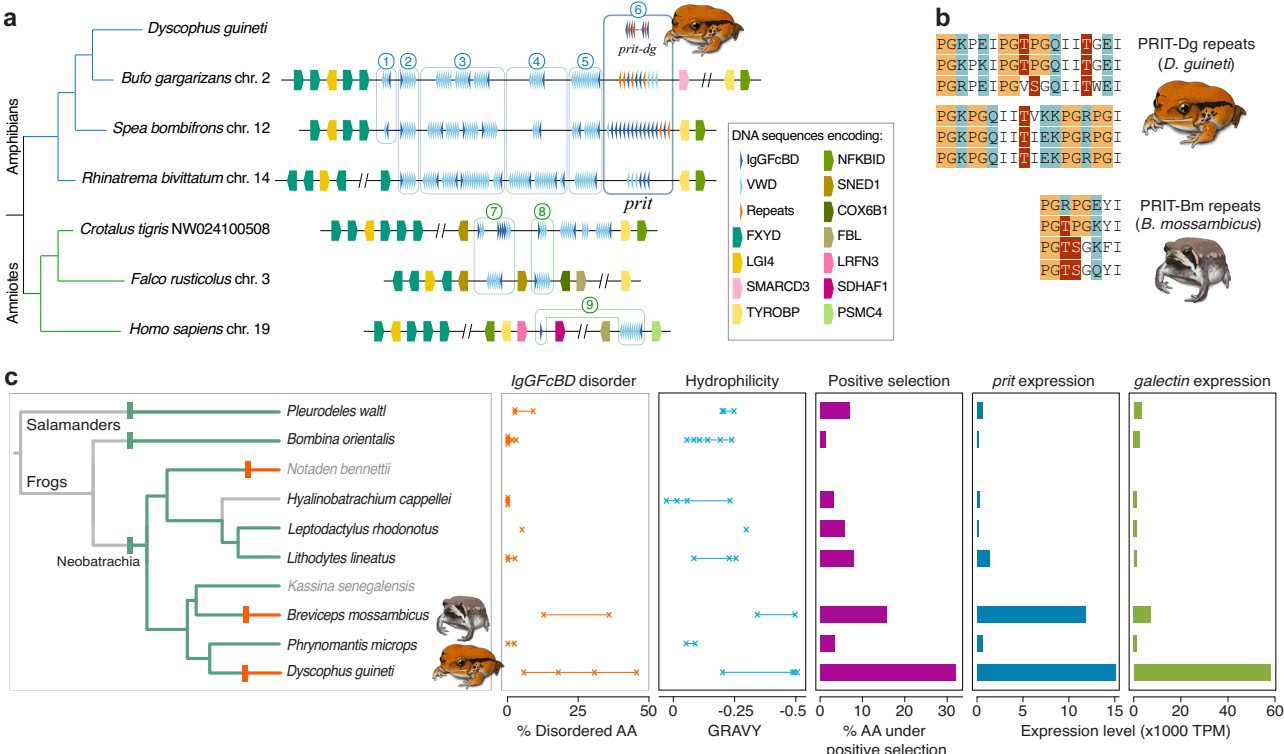

**Fig. 3 | Amphibian glues arose by structural as well as regulatory changes.**
**a** Schematic comparison of an evolutionary conserved gene cluster encoding IgGFcBD-containing proteins across a selection of screened vertebrates. Chromosomes or genome scaffolds are indicated after the species name. Triangles indicate the relative positions and orientation of IgGFcBD-encoding exons (dark blue), VWD-encoding exons (bright blue) and interdomain repeats (red). Blue windows represent gene lineages as inferred by phylogenetic analyses, with numbered gene clusters corresponding to the associated clades within the IgGFcBD tree (Supplementary Fig. S9). The gene lineage encoding PRITs is defined by a darker window. **b** Comparison of six repeats in the PRITs of *D. guineti* (three repeats from its N-terminal region and three repeats from its C-terminal region) and another glue-secreting species, *Breviceps mossambicus* (four repeats). PG motifs, glycosylation sites and polar residues are coloured orange, dark red and teal, respectively. **c** The parallel evolution of defence glues in *D. guineti* and *B. mossambicus* (mapped here on a simplified species tree) is reflected by increased disorder and hydrophilicity in the IgGFcBD of PRITs, an increased proportion of sites under positive selection in these domains, and elevated expression of both PRIT and galectin genes. Glue-producing lineages are depicted as red branches in the tree, whereas lineages secreting nonadhesive poisons are shown as green branches. Vertical red bars on tree branches denote evolutionary transitions from poison to glue; green bars represent the latest possible origins of poisons, as indicated by the sharing of homologous toxin peptides[11].

vertebrates, most IgGFcBD-containing proteins are encoded by a single gene cluster whose synteny with flanking genes is evolutionarily conserved (Fig. 3a). Remarkably, despite the presence of this cluster in all screened vertebrates, none of its genes has been functionally annotated in any taxon.

Phylogenetic analyses of IgGFcBD sequences extracted from these transcriptomes and genomes indicate that PRIT-Dg, along with other amphibian skin-expressed proteins, represent a gene lineage that diverged from other genes in this cluster in an early amphibian ancestor (Fig. 3a and Supplementary Fig. 9). Unlike PRIT-Dg, the majority of genes in this cluster encode only a single IgGFcBD followed by one or more von Willebrand D (VWD) supradomains (composed of VWD, C-8 and Trypsin Inhibitory Loop domains) that are shared with mucins and other extracellular matrix glycoproteins (e.g., IgGFcBP, sco-spondins, tectorin alpha and zonadhesins)[24]. However, the gene lineage that gave rise to PRIT-Dg and other skin-expressed proteins (hereafter all referred to as PRITs) underwent major remodelling by: (*i*) loss of VWD supradomains (testified by their absence in some caecilian and frog proteins, including PRIT-Dg), (*ii*) frequent exon duplication yielding multiple tandem-arranged IgGFcBD, and (*iii*) parallel origins of interdomain repeats in several amphibian taxa. We found distinct repeats in the caecilian *Geotrypetes seraphini* and in various frogs, including *Phrynomantis microps* (a microhylid relative of *D. guineti*) and, notably, in the glue-producing *B. mossambicus*. The repeats in *P. microps* PRIT are very similar to those of PRIT-Dg, implying a common ancestry, whereas those in *B. mossambicus* PRIT (PRIT-Bm) may either represent an independent origin or strong divergence after a common origin. The IgGFcBD-flanking repeats in PRIT-Bm are shorter than those of *D. guineti* PRIT-Dg (only eight residues long, as opposed to 18) but show similar PG dipeptide motifs and either a serine or threonine residue predicted to serve as a GalNAc-type *O*-linked glycosylation site (Fig. 3b).

To further identify any derived features that may distinguish PRIT-Dg from its homologues in nonadhesive skin secretions, we compared IgGFcBD structures across different amphibian species. Secondary structure predictions indicate that all IgGFcBD contain a similarly large proportion of β-sheets, with a median ≈ 31% of residues: a proportion that appears to be maintained in glue proteins, with no discernible shift. However, in line with their flanking repeats, the IgGFcBD of PRIT-Dg and PRIT-Bm both show a tendency towards increased disorder compared to homologues in nonadhesive skin secretions (disorder ranging from 5.6% to 30.8% of residues in PRIT-Dg IgGFcBD and 12.7 to 36% of residues in PRIT-Bm IgGFcBD, in contrast with a range of 0% to 9% in the IgGFcBD of nonglue PRITs; Fig. 3c). Furthermore, domains in both glues show more negative grand average of hydropathy[31] (GRAVY) values than their counterparts in nonadhesive secretions (Fig. 3c), implying increased hydrophilicity and thus increased solubility in water. To investigate whether these parallel shifts towards increased disorder and hydrophilicity represent adaptive evolution, we searched for evidence of positive selection by estimating ratios of nonsynonymous over synonymous codon substitutions[32] of all IgGFcBD found in PRITs. These analyses uncovered 127 (31.9 %) and 62 (15.6 %) codons that are likely to be under positive selection within the IgGFcBD of PRIT-Dg and PRIT-Bm, respectively. In contrast, equivalent values in species with nonadhesive secretions range from five (1.3% in *B. orientalis*) to 33 codons (8.3% in the salamander *Pleurodeles waltl*; Fig. 3c). Instead of defining a specific region, the residues encoded by these codons are scattered across the domain's length. This result is consistent with adaptive evolution acting on the domain as a whole to change a general property, rather than optimising specific parts.

Similar to PRITs, galectins were identified in all examined amphibian skin transcriptomes, confirming an early evolved skin function for these proteins as well. Phylogenetic analyses place both galectin-Dg1 and galectin-Dg2 as a sister-clade of galectin-9 proteins, along with various amphibian skin-expressed proteins and amniote proteins previously annotated as 'galectin-9 like' (Supplementary Fig. 10). Although the majority of proteins within the galectin-9 clade and its sister clade have retained the two domains characteristic of bi-CRD galectins (i.e., galectins with two covalently bonded CRD), the two *D. guineti* proteins are recovered together with mono-CRD galectins of another microhylid, *P. microps*, implying a relatively recent domain loss. Interestingly, the highest expressed galectin in *B. mossambicus*, galectin-Bm1, is orthologous to galectin-1 of other vertebrates, which is a canonical mono-CRD galectin. It therefore appears that glue-producing species independently recruited different members of the galectin family into their skin secretions.

Besides examining the structural features that define glue proteins, we also compared gene expression levels of PRITs and galectins across amphibian skin libraries. This comparison highlights a prominent pattern that is strongly correlated with the taxonomic distribution of glues: both *D. guineti* and *B. mossambicus* show elevated expression of PRITs in their skin (Fig. 3c). The expression level of PRIT-Dg is estimated to be 11 to 119 times higher than those of its highest-expressed homologues in six species with nonadhesive secretions, while that of PRIT-Bm is estimated to be eight to 94 times higher. Similarly, the expression of galectins is on average 17 to 50 times higher in *D. guineti* and two to six times higher in *B. mossambicus*, relative to nonadhesive species. These ratios deviate significantly from those expected under stochastic interspecific variation, as evidenced by a ratio distribution inferred from 200 randomly selected single-copy genes ($p < 0.006$ in all cases; Supplementary Data 1). We conclude that, in addition to structural modifications, changes in gene expression played a crucial role in the parallel evolution of adhesive skin secretions in amphibians.

## Discussion

The present study elucidates the molecular basis of an amphibian adhesive defence secretion by characterising two of its most abundantly expressed protein families and providing evidence for their interaction in the glue mass. Despite serving different purposes, several of the structural traits observed in glue-specific PRITs are analogous to those reported for the adhesive proteins of other animals. These features include: (1) the modular nature of the proteins (described for sea star adhesive footprints[5]); (2) glycine- and proline-rich repeat sequences (also present in spider flagelliform silk[12] and mussel byssal threads[33]); (3) a high degree of intrinsic disorder (similar to barnacle cement[34] and velvet worm slime[35]); (4) abundant glycosylation sites (such as in silkworm proteins[36]); and (5) β-sheet dominated protein domains (prevalent in spider dragline silk[37]). Interestingly, most of these traits (with the exception of elevated intrinsic disorder) also characterise at least some PRITs identified in amphibians lacking adhesive skin secretions, implying that several structural traits typically associated with adhesive proteins evolved prior to the origin of glues in *D. guineti* and *B. mossambicus* (see further). By interpreting our findings in light of the well-studied adhesives of other animals, we propose a model for the functioning of amphibian glue and its components (Fig. 4). This model relies on three interrelated mechanical properties that emerge from inferred structural features and interactions: *(1)* tensile strength; *(2)* flexibility; and *(3)* extensibility.

First, we postulate that upon secretion of granules containing glue proteins from the frog's skin, mechanical stress (e.g., pressure from a predator's bite) triggers the release of these proteins and their aggregation to form a supramolecular mesh (i.e., the glue matrix). The cohesive and adhesive strengths of this secreted glue may be determined by various multivalent interactions. Adhesive strength is most likely provided by the PRIT's glycans forming hydrogen bonds with foreign substrates - for example, the oral epithelia of a predator. This mechanism of adhesion would be comparable to the glycoprotein-mediated glue secreted by spiders to coat silk threads for prey capture[38], and more generally, to glycoprotein-based cell-cell

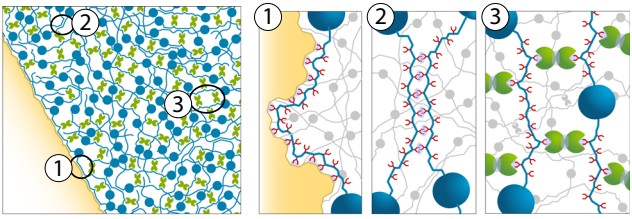

**Fig. 4 | Proposed model for the functioning of *D. guineti* glue.** Tensile strength is determined by various multivalent interactions between the major glue components, PRIT-Dg (blue) and galectin-Dg1 (green). The highly disordered interdomain repeats of PRIT-Dg result in a very flexible, conformationally adaptable protein capable of compacting under pressure to fold along the surface of a foreign substrate (yellow). Adhesive strength is provided by the formation of hydrogen bonds (purple) between the glycan side branches of PRIT-Dg (red) *(1)*, while cohesive strength is conferred by multivalent interactions between the glycan side branches of adjacent PRIT-Dg polypeptide chains *(2)* as well as the cross-linking of these glycans with galectin-Dg1 *(3)*.

adhesion[39]. Cohesive strength may result from noncovalent bonds (between adjacent PRIT polypeptide chains and glycans) and, perhaps most importantly, by the multivalent cross-linking of glycans (enabled by the highly repetitive nature of PRIT) with galectins to form a network. We hypothesise that this galectin cross-linking may require more curing time but eventually leads to stronger cohesion, explaining our observation that adhesive strength (solely relying on H-bonds) increased faster but was ultimately surpassed by cohesive strength. Relative to *D. guineti*, the lower expression level of galectin in *B. mossambicus* skin may be suggestive of either weaker cohesive strength, stronger or more effective binding of these galectins, or the potential involvement of other, as-yet unidentified components.

Second, the highly disordered regions of PRITs, mostly composed of interdomain repeats, make for a very flexible protein that is conformationally adaptable to varied surfaces. This flexibility may generate a pressure sensitivity: applied pressure, such as that caused by a predator biting down on the frog, not only compacts the glue mass to increase interaction between its constituents (and thus its cohesive strength) but also forces PRITs to fold along the substrate's surface, maximising its contact area with the substrate.

Third, similar to the disordered regions in spider silk proteins[40], interdomain repeats may also provide extensibility to PRITs by the breaking of intrachain hydrogen bonds under tensile strain. For a given tensile strength, extensibility increases the energy required to break a glue. The high extensibility of spider silk proteins, for example, has been attributed to the unfolding of β-turns, β-spirals and $3_{10}$-helices in the proteins' disordered regions characterised by high glycine and proline contents[41]. As PG is one of the most frequently encountered dipeptide motifs in β-turns[42], it seems plausible that the tandem PG motifs observed in the interdomain repeats of PRITs in *D. guineti* and *B. mossambicus* sustain such secondary structures as well.

We envisage two ways through which the β-sheet-rich IgGFcBD of PRITs may contribute to the glue's functioning. On the one hand, IgGFcBD may enhance the glue's strength by adding structural rigidity, similar to crystalline β-sheet domains in spider dragline silk and silkworm cocoon fibres. On the other hand, the modular nature of PRITs with alternating folded (IgGFcBD) and disordered (interdomain repeat) regions may underlie a stepwise extension process that further enhances the energy required to break the glue. Under tensile strain, initial extension of such proteins happens by stretching of their disordered regions. However, further extension may be achieved by the (partial) unfolding of IgGFcBD before full extension is reached, at which point the protein starts detaching from its surroundings (including substrate, other PRITs and galectins; i.e., breaking the glue). If the force required to unfold and extend a single IgGFcBD exceeds

the force required to detach a PRIT protein from its surroundings, the domains limit a single protein's extensibility but may add to the glue's strength. Conversely, if the force required to unfold a domain is below the detachment threshold, it adds to the protein's extensibility by providing so-called sacrificial bonds[43], and hence increases the toughness of the glue. The elevated disorder observed here for PRIT IgGFcBD relative to those of other proteins may very well be an adaptation to facilitate unfolding, supporting the second scenario. Analyses of an abalone nacre adhesive demonstrated that extension of a modular protein with multiple domains supported by weak bonds requires considerably more energy to reach its breaking point than a nonmodular one with the same tensile strength[44]. By maximising energy expenditure rather than simply tensile strength, the presence of multiple IgGFcBD in PRITs may increase the value of amphibian glue as a defence system adapted to exhaust a predator.

A remaining question regarding the functioning of defence glues in *D. guineti* is how it is prevented from solidifying within the skin glands. Our results suggest that the secretion of intact granules followed by their rupture under pressure may play a key role in its activation. However, we anticipate that additional factors, such as shearing[45], water evaporation or changes in physical conditions between the granules and the external environment may trigger the formation of a supramolecular structure. Functional experiments based on recombinant constructs could be used to investigate this idea, as well as evaluate the potential of PRIT for the development of bio-inspired adhesives.

Our evolutionary analyses reveal how adhesive secretions arose from a nonadhesive predecessor, highlighting them as a tractable case study for recurrent evolutionary innovation at the molecular scale. The process of structural and regulatory changes that gave rise to amphibian glues fits a unifying model for evolutionary innovation in which three successive phases – potentiation, actualisation and refinement – can be defined[46]. First, the evolution of glue was potentiated by the stepwise transformation of an ancestral mucin-like protein[24] into a suitable adhesive for a defence function. This process involved exon duplication to obtain a modular protein and the origin of interdomain repeats *prior to* the actual origin of glues. Indeed, the finding of homologous repeats and multiple IgGFcBD in skin-secreted proteins of *P. microps* implies that similar proteins with a modular architecture already existed in an ancestor shared with *D. guineti*. This indicates that the dynamic evolution of PRITs in amphibians was originally not driven by natural selection to optimise its function in glue. If so, the eventual recruitment of PRITs as glue proteins represents a case of recurrent molecular exaptation, after they originally evolved to serve another skin-related function.

Besides specific functions (e.g., the previously described isomerase in *Bombina* frogs[25]), one possibility is that PRITs in ancestral and modern species without glues sustain a similar cross-linking mechanism together with galectins. By forming a loose matrix, such a mechanism could increase the viscosity of a poison, causing it to behave like a gel to improve its contact with a predator's oral epithelia and thus enhance toxin absorption[47]. At a biochemical scale, this proposed function is closely related to an adhesive one, facilitating the later recruitment of cross-linking proteins as glue components. Second, although structural changes generated suitable glue proteins, the eventual advent of adhesive skin secretions in the respective ancestors of *D. guineti* and *B. mossambicus* was actuated by regulatory changes. The mechanisms causing elevated expression of PRIT and galectin genes are currently unknown and may have involved changes to their regulatory regions (or any gene involved in their transcriptional regulation), epigenetic processes or posttranscriptional regulation. Third, once initial increases in gene expression yielded rudimentary glues, these became subject to natural selection. Adaptive mutations refined their functioning by further optimising protein structures and possibly expression levels. The dispersed patterns of positive selection in the

IgGFcBD of both *D. guineti* and *B. mossambicus* PRITs are plausible examples of such refinement, as an overall increase in their disorder may reflect an adaptive adjustment of the force requirements for their unfolding during protein extension.

*D. guineti* and *B. mossambicus* belong to distinct neobatrachian radiations of frogs (Microhylidae and Afrobatrachia, respectively) in which many species produce nonadhesive poisons[11]. In addition, species of both radiations, like *P. microps* (Microhylidae) and *Kassina senegalensis* (Afrobatrachia), secrete toxins of the so-called 'frog skin active peptide' family (Interpro: IPR016322), which originated in an early neobatrachian ancestor[11,48,49]. Consequently, there is little doubt that *D. guineti* and *B. mossambicus* descended from a poisonous peptide-secreting ancestor and that their glues evolved independently. Our study shows that this parallel evolution extends down to striking molecular detail. Interestingly, the adhesive protein Nb1 of the Australian frog *N. bennetti* has also been reported to contain two IgGFcBD and repeats[25] and may represent a third independent origin of a similar glue protein.

Recurrently evolved adaptations in the history of Life are interesting because they demonstrate the degree of similarity that can be expected when different organisms are exposed to similar selective pressures. In the case of amphibian glues, we can conceive two factors that rendered the likelihood of their parallel evolution much higher than one might expect: (1) the widespread availability of a suitable protein template for a glue to evolve; and (2) the multitude of mutations through which increased protein disorder, increased hydrophilicity and elevated gene expression can be achieved.

The evolutionarily conserved expression of PRITs in amphibian skins increased the chance of their repurposing to glue proteins in multiple lineages, especially if they already served a related function as postulated above. At a broader phylogenetic scale, both mucin-like glycoproteins and lectins have been co-opted into the adhesive secretions of a wide range of animals[50,51]. Unlike other extracellular domains (including those of the VWD supradomain), IgGFcBD is currently not known as a frequent structure in animal adhesives[50], despite lending its name to a widespread family of extracellular proteins. Its parallel inclusion in the glues of three frog lineages could perhaps change this, and we anticipate that other glue-secreting amphibians (or perhaps other animals) may have evolved adhesives by incorporating the same domain. In this respect, amphibian adhesives add to an emerging paradigm that identifies extracellular proteins involved in supramolecular structures as an ancient, universally available recruitment ground for biological glues.

## Methods

### Ethics statement
Experiments involving live frogs were conducted in accordance with European guidelines and Belgian legislation on animal housing and experimentation. All procedures were approved by the Ethical Committee of Animal Experimentation of the Vrije Universiteit Brussel (permit no. EC16-334-1).

### Sample collection
Adult *Dyscophus guineti* frogs (*n* = 10) were purchased from the pet trade. For transcriptome analysis, three individuals were immediately euthanised using a 10% lidocaine solution (10 μL/g body weight) and freshly dissected skin tissue was stored in RNAlater (Sigma-Aldrich; St. Louis, MO, United States). For proteome analysis, adhesive and mucus secretions from three animals were sampled separately in dH$_2$O and immediately flash-frozen for storage at -80°C. For lectin staining, SEM and immunolabeling, glue and mucus were sampled by either manually massaging the animals to induce skin secretion or through mild electrostimulation of skin glands (max. 2 V). Glue was collected on either clean glass coverslips (SEM) or microscope slides (lectin staining and immunolabeling) by lightly touching the animal's dorsum

(yielding images after applying low pressure to the glue) or by applying manual pressure (yielding images after applying moderate pressure to the glue). For functional tests, we collected glue from *D. guineti* (*n* = 7) as well as the poisonous skin secretion of *B. orientalis* (*n* = 3) on one of the smooth sides of 2 × 2 LEGO® bricks (see below). For SEM, glue and mucus were fixed in Bouin's solution (Sigma-Aldrich) for 24 h. For all other applications, slides were fixed in 4% (wt/vol) paraformaldehyde (PFA) in sodium phosphate buffer (PBS solution, pH 7.4). For histology and immunohistochemistry, skin samples were similarly fixed in Bouin's solution and dehydrated with an ethanol series. Decalcification of the dermal layer was carried out using a 1:1 mixture of 2% ascorbic acid and 0.3 M NaCl (final pH ∼ 2.6) under shaking, after which the tissues were embedded in paraffin wax and sectioned transversely using a microtome set to 5 μm.

### Mechanical pull tests
LEGO® bricks, composed of acrylonitrile butadiene styrene (The Lego Group; Billund, Denmark) were selected as adherends for mechanical pull tests due to their smooth surfaces and precise dimensions. All tests were conducted at an ambient temperature of 20–21 °C and a relative humidity of 44–45% to minimise variance[52]. For each replicate, one side of a brick (9.6 mm × 15.8 mm = 151.68 mm$^2$) was coated with skin secretion at a mass fixed between 6 and 10 mg, measured by weighing the brick immediately before and after coating. The brick was subsequently pushed against a second brick in a custom 3D-printed brick holder by manually applying a force (corresponding to a pre-specified pressure) measured with a digital dynamometer (Sauter FK100; precision 0.5 N; Metil industries, Deerlijk Belgium; Supplementary Fig. 1a), after which the glue was allowed to cure. No additional pressure was applied to the bond during curing. Mechanical pull tests were conducted by placing the dynamometer as well as the brick holder on a linear actuator controlled by a Nema 14 bipolar stepper motor (Supplementary Fig. 1b). One brick was fixed in the holder while the other was pulled by the dynamometer moving at a velocity of 25 mm/min. The dynamometer recorded the maximum pull force (N) at the time of glue failure - i.e., when the two bricks broke apart. Brick pairs underwent three different treatments: (i) joined at an applied pressure of 40 kPa and cured for 10 min (high pressure; short curing time); (ii) joined at 40 kPa and cured for 60 min (high pressure; long curing time); and (iii) joined at 4 kPa and cured for 60 min (low pressure; long curing time). Tensile strength (in kPa) was calculated by dividing the pull force (N) by the glued surface area of the bricks (151.68 mm$^2$). *B. orientalis* skin secretion failed to bond the bricks at either curing time (i.e., they remained detached), and so its tensile strength was recorded as 0 kPa.

### Protease treatment
To determine whether proteins are involved in the adhesive activity of *D. guineti* glue, tensile strength measurements were taken after treatment with one of two proteolytic enzymes: Proteinase K (Merck Corporation; Rahway, NJ, United States) or trypsin (Sigma-Aldrich). LEGO® bricks coated with fresh glue were submerged for 60 min at 37 °C in one of three different solutions: water (the solvent), proteinase K solution (2 mg/ml) and trypsin solution (2 mg/ml). At the end of this incubation period, each brick was joined to a second one in the same way as described above. Pull tests were conducted after 60 min of curing. As enzyme-treated glue failed to bond the bricks (i.e., they remained detached), its tensile strength was recorded as 0 kPa under either treatment.

### Scanning electron microscopy
After fixation in Bouin's solution, *D. guineti* glue and mucus were dehydrated in graded ethanol and dried by the critical-point method. The samples were individually mounted on aluminium stubs, following which they were coated with gold in a sputter coater and observed with a JEOL JSM 7200 F scanning electron microscope.

## Lectin histochemistry

Adhesive secretions were collected on clean microscope glass slides and fixed in 4% PFA in PBS for 30 min, then stored in PBS overnight. Glue smears were washed three times in Tris-buffered saline (TBS, pH 8.0) supplemented with 0.1% Triton (TBS-T). Nonspecific background staining was blocked by pre-incubation in TBS-T containing 3% (w/v) BSA (BSA-T) for 1 h at 4 °C. Commercially available biotinylated lectins (Con A, WGA, RCA$_{120}$, UEA I, PHA-L, PNA, SBA; Vector Laboratories; Newark, CA, United States) were diluted in BSA-T to a final concentration of 25 μg/ml and applied to the samples for 2 h at 4 °C. After three washes of 5 min each in TBS-T, the samples were incubated for 1 h in Texas Red™-conjugated streptavidin (Vector Laboratories) diluted 1:100 in BSA-T at room temperature. After three 10-min washing steps in TBS-T, the sections were mounted in Vectashield mounting medium with DAPI and analysed with a Zeiss Axioscope A1 microscope.

## Transcriptomics

Skin tissues from three frogs (150 mg per individual) were homogenised using a GentleMacsTM disassociator (Milteny Biotec; Bergisch Gladbach, Germany) and RNA was extracted using the RNeasy Universal Plus Midi kit (Qiagen; Hilden, Germany). Purified RNA of individual frogs was pooled together for RNA-seq analysis. A whole-transcriptome paired-end sequencing library was constructed, which involved the sequencing of minimum 50 million paired-end 100 (PE100) reads using a TruSeq stranded RNA-seq library preparation kit (Illumina Inc.; San Diego, CA, USA) and the Illumina HiSeqTM™ 2500 sequencing system (outsourced to BaseClear BV; Leiden, The Netherlands). After quality control using in-house protocols, raw reads were converted to FASTQ sequence files. Transcript sequences were reconstructed de novo using a pipeline involving: (i) contig assembly by Trinity 2.15.1[53] (reconstructed according to *(1)* no strand-specificity, *(2)* RF strand-specificity and *(3)* FR strand-specificity); (ii) filtering and clustering of closely related contigs across assemblies using EvidentialGene[54]; and (iii) estimation of transcript expression levels (in transcripts per million; TPM) using kallisto 0.44[55]. The skin libraries of one additional species producing glue (*Breviceps mossambicus*) and six amphibians producing nonadhesive skin secretions (*Pleurodeles waltl*, *Bombina orientalis*, *Hyalinobatrachium cappellei*, *Leptodactylus rhodonotus*, *Lithodytes lineatus* and *Phrynomantis microps*) were similarly assembled using this approach. The completeness of each transcriptome assembly was assessed using the BUSCO[56] metric with a dataset limited to vertebrates, which resulted in the recovery of between 69.1% and 82.1% near-universal single copy orthologs.

## Mass spectrometry

To allow robust protein identification and quantification, two secretion types (glue and mucus) were sampled from each of three *D. guineti* individuals. Proteins were purified from these samples using reversed-phase adsorbent Sep-Pak C8 Plus cartridges (Waters; Antwerp, Belgium). Lyophilised eluates were dissolved in 8 M urea - 5 mM DTT- 30 Mm Tris buffer and their protein contents were digested with trypsin (Thermo Scientific; Waltham, MA, United States). The resulting peptides were desalted using Pierce C18 spin columns (Thermo Scientific) and fractionated in an Ultimate 3000 UPLC system (Thermo Scientific), followed by a Q Exactive Orbitrap mass spectrometer (Thermo Scientific). Mass data were acquired with Xcalibur 3.1.66.10 software (Thermo Scientific). Sequenced peptides and mass spectra (PRIDE: PXD045803) were screened using MASCOT version 2.2.06 (Matrix Science; London, United Kingdom) against the *D. guineti* dorsal skin transcriptome as a database. Scaffold 3.6.5 (Proteome Software; Portland, OR, United States) was used to determine the false discovery rate (FDR) and evaluate protein inference, while protein quantification was calculated in Progenesis version 4.0 (Waters) based on the normalised abundance of all matching features.

## Screening and identification of glue candidates

The obtained *D. guineti* skin library was screened using BLAST homology searches against public databases filtered for vertebrate sequences. Only transcripts with TPM values ≥ 2.0 were screened so as to minimise the contribution of contaminants or assembly artifacts. Potential hits were used as queries against the Conserved Domains Database (CDD) of NCBI[57] and InterPro[58] with default settings. To guide the selection of potential candidates, transcripts from the dorsal skin library were first cross-referenced with data from MS analysis of the skin secretions. Sequences recovered using both methods were then ranked according to their TPM values (transcripts) and abundance of corresponding proteins in the glue. Two proteins, PRIT-Dg and galectin-Dg1, were found to be prominent by being highly ranked in both lists. All possible homologues of PRIT-Dg and galectin-Dg1 were subsequently identified in skin libraries of seven additional amphibian species (listed above under "Transcriptomics") using blastx searches. To ensure the comparability of TPM values across skin transcriptome libraries of the eight different species, expression levels of 200 single-copy genes that are not expected to be differentially expressed were evaluated (Supplementary Data 1). This dataset included six housekeeping genes (calnexin, cytochrome c1, beta-glucuronidase, hypoxanthine-guanine phosphoribosyltransferase, succinate dehydrogenase complex subunit A, TATA box binding protein) and 194 randomly selected BUSCOs shared by all eight libraries. For each of these 200 genes, we calculated the ratio $TPM_{glue}$ / $TPM_{nonglue}$, where $TPM_{glue}$ is its expression level in either *D. guineti* or *B. mossambicus* and $TPM_{nonglue}$ is its maximum expression level among the six species with nonadhesive secretions. The distributions of these ratios (200 per distribution) were used to evaluate the probability that the corresponding ratios for PRIT and galectin may be attributed to stochastic variation in gene expression.

## Amplification of full-length transcripts

To obtain full-length sequences of the two glue-encoding transcripts, aliquots of 10-μg and 1-μg (for 5' and 3' RACE, respectively) of total RNA were reverse-transcribed using the FirstChoice RLM-RACE kit (Invitrogen; Waltham, MA, United States). Degenerate primers for PRIT-Dg were designed based on two criteria: (1) presence of suitable primer binding sites in their underlying transcripts; and (2) coverage of these regions by sequenced peptides in the glue proteome. These custom primers were used in combination with the 5'RACE and 3'RACE outer and inner primers (Invitrogen) to amplify DNA fragments from the cDNA obtained using the 5'- and 3'-RACE protocols of the RLM-RACE kit, respectively. PCRs were carried out using thermocycling conditions and parameters described in the kit protocol. Amplified DNA fragments were excised and purified using QIAquick PCR and Gel Extraction Kit (Qiagen) and subsequently cloned into the pCR2.1-TOPO vector using the TOPO TA Cloning kit (Invitrogen) before transformation into OneShot Top10 chemically competent *Escherichia coli* cells (Invitrogen). A minimum of 10 clones were selected for lysate PCR and used to verify DNA insertion, and clones containing correctly sized fragments were submitted for Sanger sequencing (BaseClear). Sequences were checked manually and aligned using CodonCode Aligner 10.0.2 (CodonCode Corporation; Dedham, MA, USA) and MAFFT 7.0[59]. Additional sense and antisense primers were designed based on the newly acquired sequences in an iterative approach and used to amplify inwards. Additionally, for PRIT-Dg, total RNA was amplified with the qScript Flex cDNA kit (QuantaBio; Beverly, MA, United States) using oligo-dT primers for synthesis of long cDNA sequences. The polymerase AccuStart Long Range SuperMix (QuantaBio) was then used in conjunction with primers located near the 5'- and 3'- terminals, as determined by RACE-PCR. The resulting amplicons were submitted for primer walking (outsourced to BaseClear); however, due to the high sequence similarity and variable numbers of

interdomain repeats, complete amplicon sequences could not be recovered.

## SDS-PAGE and western blotting

Protein content in *D. guineti* glue and mucus was visualised using SDS-PAGE, with samples run under both nonreducing and reducing (i.e., with added dithiothreitol, DTT) conditions (Supplementary Fig. 2). Due to the low protein concentration in mucus relative to the highly proteinaceous glue (e.g., ~ 0.1 mg/ml and upwards of 5 mg/ml, respectively, based on Nanodrop estimations), glue samples were diluted to approximate the total protein concentration of mucus and the gel stained with the Pierce™ Silver Stain Kit (Thermo Scientific). Samples were loaded alongside the PageRuler™ Prestained Protein Ladder and HiMark™ Pre-stained Protein Standard (Thermo Scientific) as references for the estimation of molecular weights ranging from 10 to 180 kDa and 30 to 460 kDa, respectively. Based on sequences of PRIT-Dg and galectin-Dg1 acquired from transcriptome and proteome data, antigenic peptides were identified, synthesised and used to raise polyclonal antibodies for each glue protein (outsourced to Eurogentec; Seraing, Belgium). Antibodies were raised against the following targets: (1) an IgGFcBD in PRIT-Dg (QANFKKEMKVRKGQT); (2) the repeat region in between two successive IgGFcBD of PRIT-Dg (QII-TEEIPGRPEIPG); and (3) galectin-Dg1 (GPGDNFEVEIRNEG). Antibody specificity was verified using Western blots. Briefly, proteins were denatured in SDS and boiled for 5 min, then resolved on a 4-15% (wt/vol) Mini-PROTEAN TGX precast polyacrylamide gel (Bio-Rad; Hercules, CA, United States). After electrophoresis, proteins were transferred using the Trans-Blot Turbo Mini 0.2 µm PVDF Transfer Pack together with the rapid mini-gel protocol of the Trans-Blot Turbo Transfer System (Bio-Rad). The blots were immunodetected using standard procedures: in short, they were probed overnight with the three sets of purified polyclonal antibodies at dilutions of 1:1,000, followed by anti-rabbit IgG, HRP-linked antibodies (Cell Signalling Technology; Danvers, MA, United States) diluted 1:2,000 and, finally, chemiluminescence detection (GE Healthcare; Chicago, IL, United States). For PRIT-Dg, antibodies targeting the IgGFcBD domain and the interdomain repeats each resulted in identical banding patterns (Supplementary Fig. 11), due to which only the latter was prioritised for subsequent applications. To assess whether PRIT-Dg undergoes *N*- and *O*-glycosylation, the majority of linked glycans were removed by treating the secretion with Protein Deglycosylation Mix II (NEB; Ipswich, MA, United States) using the kit's denaturing reaction protocol and an overnight incubation step, followed by immunoblotting against PRIT-Dg as described above.

## Protein-protein interactions

To determine whether protein-protein interactions between PRIT-Dg and galectin-Dg1 occur within the glue milieu, two protocols were followed: 1) co-immunoprecipitation, and 2) native PAGE (i.e., non-denaturing protein separation). For co-immunoprecipitation, the Immunoprecipitation kit (Abcam; Cambridge, United Kingdom) was used with modifications to the kit protocol to minimise nonspecific "sticky" binding: glue containing approximately 0.5 µg of total protein was pre-cleared with Protein A/G Sepharose® beads (previously blocked using 1% BSA in PBS for one hour at 4ºC) prior to proceeding with antibody binding and bead capture. Immunoprecipitation was carried out with both anti-PRIT-Dg and anti-galectin-Dg1 antibodies to verify whether both proteins were simultaneously recovered as a protein-protein complex. For native PAGE, 4% native polyacrylamide gels were prepared from 30% acrylamide/bis-acrylamide stock (Severn Biotech Limited; Worcestershire, United Kingdom) with 0.1% TEMED, 0.032% APS and 10 mM HEPES (pH 7.5), after which native PAGE loading dye (62.5 mM Tris-HCl, pH 6.8, 25% glycerol, 1% bromophenol blue) was added to the samples at 1X concentration. Since PRIT-Dg has a theoretical isoelectric point of 9.6, we expect it to remain basic even

in a complex with galectin-Dg1, which has a theoretical isoelectric point of 5.1; accordingly, gels were run in 10 mM HEPES (pH 7.5) running buffer at 4 °C with reversed electrodes. Since native PAGE separation is based on both protein mass and charge, there is no linear relationship between mass and migration (as in SDS-PAGE), and thus no molecular weight ladder was included. Proteins were transferred from the native PAGE gel to a PVDF membrane using the Trans-Blot Turbo Transfer system (Bio-Rad). Immunoblots were subsequently carried out with specific anti-PRIT-Dg and anti-galectin-Dg1 antibodies, with membrane stripping performed in between blots to ensure replicability of results. Membranes were stripped by incubation with a buffer containing 200 mM glycine, 3.5 mM SDS and 1% Tween-20 at pH 2.2 for 20 min at room temperature.

## Immunolabelling

Glue and mucus secretions fixed in PFA and stored in PBS overnight were dehydrated through an ethanol series and, after one wash of 3 min in water, blocked for several hours in PBS solution containing 0.05% (vol/vol) Triton and 3% (wt/vol) BSA (PBS-T-BSA). Antibodies against PRIT-Dg and galectin-Dg1 were diluted 1:100 in PBS-T-BSA and applied to the sections overnight at 4 °C, followed by several washing steps in PBS-T and incubation for 1 h in either Alexa Fluor 488-conjugated anti-rabbit immunoglobulins (Invitrogen) diluted 1:250 (for PRIT-Dg) or Fluorescein (FITC)-conjugated anti-rabbit immunoglobulins (Proteintech Group; Rosemont, IL, United States) diluted 1:100 (for galectin-Dg1). The secretions were mounted with Vecta-shield (Vector Laboratories) and observed using a Zeiss Axioscope A1 microscope.

## Histology and immunohistochemistry

One section per skin type (i.e., dorsal or limb tissue, fixed in Bouin's solution) was stained with Heidenhain azan trichrome[60], while the others were submitted to the same staining method as the whole secretions but with the addition of an antigen retrieval step. This was achieved by incubation in a solution containing 0.05% (wt/vol) trypsin (Sigma-Aldrich) and 0.1% (wt/vol) CaCl$_2$ for 15 min at 37 °C. Sections were observed in the same manner as the secretions, described above.

## Structural analyses

The NetNGlyc 1.0[61] and NetOGlyc 4.0[62] tools were used to predict *N*-linked and *O*-GalNAc glycosylation sites, respectively, along the PRIT-Dg protein, with a score ≥ 0.5 for a site implying that it is more likely to be glycosylated than not. Intrinsic disorder in interdomain repeat regions and IgGFcBD was estimated with IUPred3[63]. To compare degrees of disorder in IgGFcBD of eight amphibians (Fig. 3c), we inferred for each domain the percentage of its amino acids likely to be part of a disordered region (reflected by a site-specific score of ≥ 0.5). Similarly, to compare hydrophilicity across these species, we calculated the grand average of hydropathy (GRAVY) for each of their IgGFcBD using the online ProtParam tool[64]. IgGFcBD secondary structures were predicted with GOR 4[65], and structural predictions for a segment of PRIT-Dg containing one IgGFcBD and its flanking regions were obtained with AlphaFold2 2.1.1[27] using the monomer_ptm model to obtain a per-residue confidence measure (pLDDT).

## Genome screening and phylogenetic analysis

A total of 16 vertebrate genomes were screened to investigate the diversity and organisation of IgGFcBD-containing genes. Screening was conducted using tBLASTn searches against the NCBI Genome Data Viewer with previously retrieved IgGFcBD and VWD supradomains as query sequences. Sequences of 167 IgGFcBD-containing proteins (Supplementary Data 2) and 179 galectins (Supplementary Data 3) were retrieved and aligned using the E-INS-I algorithm in MAFFT 7. Phylogenetic relationships were estimated for both datasets by Bayesian inference and maximum likelihood (ML) using MrBayes 3.2.7[66] and

RAxML[67], respectively. ModelTest-NG v0.1.7[68] defined the WAG empirical model for amino-acid substitution with gamma correction for among-site rate hereogeneity (+ G) and an estimated proportion of invariable sites (+ I) as best fitting for the IgGFcBD dataset and the LG model with gamma-correction and an estimated proportion of invariable sites (LG + G + I) as best fitting for the galectin dataset. For the Bayesian analyses, we applied a model that implemented ModelTest-NG's recommended rate heterogeneity parameters (+ G + I) but used a mixed prior for the amino-acid substitution model. Two parallel runs of four incrementally heated Markov Chain Monte Carlo (MCMC) chains (temperature parameter = 0.2) were executed for a length of 20,000,000 generations with a sampling interval of 2000 generations. Convergence of the parallel runs was confirmed by split frequency standard deviations (< 0.05) and potential scale reduction factors (approximating 1.0) for all model parameters, as reported by MrBayes. After discarding the equivalent of 25% of sampled trees, a consensus phylogram posterior probabilities for its branches were inferred from the last 15,000 sampled trees of both runs. For ML analyses, we implemented the models recommended by ModelTest-NG and inferred branch support by executing 1,000 replicates of RAxML's rapid bootstrapping algorithm.

### Selection analyses

To investigate patterns of Darwinian selection acting within the IgGFcBD of PRITs, we compiled a dataset encompassing all domain sequences of the eight amphibian species included in our comparative analysis (Fig. 3c). Codon sequences corresponding to single domains were retrieved from their transcript sequences and aligned on the basis of their corresponding amino acids using TranslatorX[69]. The resulting codon dataset, consisting of 25 codon sequences, was used as input in the program CodeML in the pamlX software package[70] along with a corresponding subtree pruned from the Bayesian consensus tree (Supplementary Fig. 9). We used a branch-site model to estimate ω values (ratios of nonsynonymous over synonymous substitutions), involving the comparison of a specified set of foreground branches (in which a proportion of codons may have evolved under positive selection) vs. the remaining background branches (in which codons are assumed to evolve under negative selection or neutral drift). For each of the eight species, a separate branch-site analysis was conducted, in which the foreground branch set included all branches that uniquely gave rise to the domain sequences of the species in question. For each foreground branch set (and thus each species), the proportion of codon sites estimated to evolve under positive selection was used as a comparative measure of positive selection among the eight species.

### Statistics and reproducibility

For mechanical pull tests, series of tensile strength values representing different glue treatments were analysed using R version 4.3.3 (https://www.r-project.org). To model the effect of the treatments on tensile strength, a general linear mixed model (LMM) was constructed using the R package lme4, including frog identity (the frog individual from which the sample was taken) as a random factor and sampled glue mass as a fixed covariate. Differences between treatments (10 min curing vs. 60 min curing; bricks glued at 40 kPa vs. bricks glued at 4 kPa; incubation of glue in water vs. without water) were tested by specifying and comparing contrasts in the model with the R package lsmeans (two-sided z-tests, probabilities not corrected for multiple comparisons). To ensure reproducibility of results, all experiments were repeated at least three times with similar outcomes, including micrographs (SEM, antibody labelling of secretions and skin tissues, histological stains) and immunoblots (Western blots for antibody specificity, deglycosylation treatment, detection of protein-protein interactions).

### Reporting summary

Further information on research design is available in the Nature Portfolio Reporting Summary linked to this article.

### Data availability

The nucleotide and protein sequences generated in this study have been deposited in the GenBank database under accession codes OR483821 (PRIT-Dg), OR480786 (galectin-Dg1), OR480787 (galectin-Dg2), OR480788 (PRIT-Bm) and OR543004 (galectin-Bm1). The mass spectrometry proteomics data have been deposited to the ProteomeXchange Consortium (http://proteomecentral.proteomexchange.org) via the PRIDE partner repository with the dataset identifier PXD045803 and https://doi.org/10.6019/PXD045803 [https://www.ebi.ac.uk/pride/archive/projects/PXD045803]. All other relevant data analysed in this study are provided in the Supplementary Information and Source Data files. Previously published accession codes cited in this study are available in the UniProt database under accession codes Q9Y6R7 (human IgGFc-binding protein) and Q58PK6 (*Bombina variegata* peptidylaminoacyl-L/D-isomerase), in the GenBank database under accession code AAX55674 (*Bombina orientalis* peptidylaminoacyl-L/D-isomerase), and in the InterPro database under accession codes IPR035234 (IgGFc-binding protein, N-terminal), IPR044156 (galectin-like protein family) and IPR016322 (frog skin active peptide family). Source data are provided with this paper.

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

## Acknowledgements

We thank Sunita Janssenswillen for assisting with protein extraction, Annelore Stroobants for assisting with gel electrophoresis, Leonard De Causmaecker for help with designing and performing the mechanical pull tests and Bram Vanschoenwinkel for help with statistics. This work is financed by FWO-Vlaanderen (grant no. G0D3214N) and Vrije Universiteit Brussel (grant no. SRP-30). S.Z., J.V.L. and G.R. are supported by doctoral fellowships from FWO-Vlaanderen (grants no. 11C6521N, 11D2520N and 1SC8222N, respectively). B.L. is supported by an ESPRIT grant of the Austrian Science Fund (FWF): [ESP 15]. P.F. is Research Director of the Fund for Scientific Research of Belgium (FRS-FNRS) and is supported by a FNRS PDR Grant no T.0088.20.

## Author contributions

K.R. conceived the project and acquired funding; S.Z., B.L., J.V.L., P.T., P.F., and K.R. designed the research; S.Z., B.L., J.V.L., G.R., L.B., I.S., S.C., and K.R. performed the research; S.Z. and K.R. analysed the data; all authors contributed to the manuscript.

## Competing interests

The authors declare no competing interests.
