## [Peer Review File · Nature Communications]

Recurrent evolution of adhesive defence systems in amphibians by parallel shifts in gene expressionREVIEWER COMMENTS

Reviewer #1 (Remarks to the Author):

The manuscript by Zaman et al. focuses on the adhesive skin secretions of the Madagascar tomato frog *Dyscophus guineti* (Microhylidae) as a model species. The authors utilized a diverse suite of methods/techniques and experimental approaches to provide direct evidence and fill-in the gaps in the current knowledge in two different directions:

- i) related to the skin-secreted glue (the anti-predator defence at the level of a frog species) – What is its composition? How the major proteins engage in complex interactions after they are released from the gland? How quickly is the adhesive formed? What are its mechanical properties?
- ii) related to the origin of the genes encoding skin glue proteins in amphibians (evolution in action involving “recurrent innovation at the molecular scale”) – What is the ancestor that gave rise to the host-defence adhesive components? What are the precise molecular mechanisms that acted on genetic level? Is this an isolated event in *D. guineti*, or it is rather a recurrent innovation that happens independently in different frog lineages?

In my opinion, this paper is suitable for publication in Nature. The work is technically sound. The methods provide enough details for the work to be reproduced. The authors raise their own specific antibodies that allows them to provide evidence and answer many of the questions related to i) above. The conclusions are supported by the results. Apart from “elucidating the molecular basis of an amphibian adhesive skin secretion”, the authors reveal for the first time “how adhesive secretions arose from a non-adhesive predecessor”. These are original findings and contribute greatly to the current knowledge in the field by filling in important missing gaps and proposing models that can be applied not only to other amphibians but also to other animals that utilize similar modular proteins with multiple domains.

Specific comments and/or questions:

1. In Discussion (p.13, l.314-318) – the authors propose that the trigger for the protein-protein bonding is the change in the physical conditions. Since this is an essential part of the presented paper, could the authors perform an experiment to confirm that?
2. In Fig. 1b - What is the rationale for illustrating only PRIT-Dg using two different antibodies in glue? Why not show in the main text the presence of galectin-Dg1 in glue and absence from mucus, instead of having it in Fig. S5? This in fact would be affirmative that the in-house antibodies are recognizing the two important proteins in the glue only.
3. In Results (p.4, l.34) – the abbreviations GalNAc and GlcNAc are mentioned for the first time. Please spell those out.
4. In Fig. S6 – Why is the pull-down experiment to analyze the co-precipitation of the two proteins is done only using anti-PRIT-Dg (interdomain repeats)? The anti-PRIT-Dg (IgGFcBD) also should be used in the pull-down step.
5. How would you account for the fact that in your proteomic analysis you were not able to detect the presence of the Kunitz family protease inhibitor as reported by Conlon and Kim (2000) in norepinephrine-stimulated skin secretions? Your only comment was that this inhibitor might be playing a role in the dorsal gland only but not in the secretions.

General comment:

1. The panels in all figures are labelled with small letters, while in the text they are referred to with capital letters. Please revise for consistency. Additionally, some of the panels are labelled on the top right, while they should all be in the top left of the panel.
2. p.21, l.497 – PRIT should PRIT-Dg.

Reviewer #2 (Remarks to the Author):

Dear authors, your work is excellent and contributes greatly to the understanding of the mechanisms involved in the adaptation of amphibians, however, some important points need to be reviewed with caution, due to delicate statements that can only be made with these results obtained.

- Between lines 44 and 48 you state that amphibians adapted in two different ways, one group that presents poison, but does not present glue, and a group that presents glue as an adaptation to the absence of poison.

I consider this statement delicate, given that many anurans, such as *Rhaebo guttatus*, *Rhinella* sp. They have poison on their skin, and they also have a poison that solidifies after the glandular secretion that also looks like glue.

<https://onlinelibrary.wiley.com/doi/epdf/10.1002/jez.1838>
<https://frontiersinzoology.biomedcentral.com/articles/10.1186/s12983-018-0294-5>

Another example is found between lines 52 and 54, where you mention hylids as glue producers, and we have important examples of toxicity in casque-headed-frogs, such as *Corythomantis grenningi* and *Aparasphenodon brunoii*, which belong to the Hylidae family.

<https://www.cambridge.org/core/journals/journal-of-zoology/article/abs/head-coossification-phragmosis-and-defence-in-the-casqueheaded-tree-frog-corythomantis-greeningi/F8C15562F3F657AAB5639A2D4AA1BE3D>

[https://www.cell.com/fulltext/S0960-9822\(15\)00788-5](https://www.cell.com/fulltext/S0960-9822(15)00788-5)

Wouldn't there be an intermediate group between the presence of glue and the presence of poison among amphibians during the adaptation process? For example, an amphibian that presents both adaptations. It's possible?

Has the presence of alkaloids or steroids been studied in *D. guineti*? Has the possibility of this group of animals producing such molecules been ruled out?

- Between lines 107 and 108 you state that the PRIT protein has multiple copies of IgGFcBD protein. How can you ensure that even divided by interdomains, they will be translated into a single molecule?

Couldn't they be translated separately? Just like what happens in the venom of snakes like the Jararaca? Which has several coding sequences in tandem, but expresses each protein individually?
<https://www.pnas.org/doi/epdf/10.1073/pnas.2015159118>

What is the predicted size of this group of proteins by the transcriptome? Is it similar to the molecular weight obtained in Wersten Blott's analyses?

- Why don't you present the electrophoresis gel of glue and mucus secretion to show the differences in the electrophoretic profile?

Why is the molecular weight standard range not shown in the Wersten Blott image?

Did you consider carrying out the electrophoresis under reducing conditions?

- Regarding proteomics, what was the purpose of performing pre-purification on C-8 before analysis? Do you believe that this type of preparation could have eliminated other important proteins such as the Kunitz-type inhibitor already described for one of the animals? Did you consider a shotgun analysis of the crude material using urea?

I strongly recommend accepting this work for publication after minor revisions

What are the noteworthy results?

Yes

Will the work be of significance to the field and related fields?

Yes

How does it compare to the established literature?

To date, no research group has managed to uncover the biochemical composition of glues produced by amphibians with this level of molecular refinement.

Does the work support the conclusions and claims, or is additional evidence needed?

Some revisions cited above

Are there any flaws in the data analysis, interpretation and conclusions? Do these prohibit publication or require revision?

Some revision cited above

Is the methodology sound?

Yes

Does the work meet the expected standards in your field?

Yes

Is there enough detail provided in the methods for the work to be reproduced?

Yes

Reviewer #3 (Remarks to the Author):

In this work by Zaman et al, the authors use transcriptomic, proteomic, and functional approaches to develop an evolutionary theory for convergent biological glues. Overall, the manuscript is well-written, the work is rigorous, and the ideas are presented clearly. This is an area of wide interest, and relating adhesive biomolecules to their evolution remains a central question in comparing diverse biological strategies. I believe this work is well-suited for this journal and recommend publication pending a few minor points related to the adhesive testing:

- Page 4, line 85: 'tensile strength' is used to describe forces observed from the pull experiments in determining adhesion. Since the authors describe their experiment to be pulling one block on a trolley in shear from one that is fixed, I believe the authors mean shear strength rather than tensile strength. Please correct this terminology.

- Page 4, line 85: The authors refer to tensile strength as 'determined by both cohesive and adhesive strength'. While this is true, it is an odd thing to point out in the text as every bond strength measurement will rely on both cohesive and adhesive forces. It is also slightly misleading as the authors did not perform separate experiments to determine cohesive versus adhesive strengths, such as determining the bulk modulus of the glue or using more sensitive surface techniques to measure adhesive forces.

- Related to the previous point, and indicated by the authors, understanding the failure mode of the adhesive can shed some light on the mechanisms that are laid out towards the end of the manuscript. This can be inferred from simple images of the samples after fracture, if they are available. If the samples are failing adhesively (glue remains on one side of the bond) then the cohesive forces are stronger than the adhesive forces in the material. On the other hand, if you observe cohesive or mixed mode failure, then the bulk forces are greater than or equal to the

adhesive force, respectively. This would be helpful in shaping the hypotheses laid out in the discussion regarding adhesive and cohesive interactions. Generally, adhesive testing should be supported by supplemental images of the samples after fracture.

- I found that the details in the adhesive testing method section were quite thin. The authors describe testing as one fixed block and another being pulled on a trolley with a strain gauge. Was the trolley pulled at a fixed velocity? Was this by hand or a screw motor? What was the rate? Was pressure applied to the bond during "curing"? Was the volume of adhesive from the frogs fixed? Many of these details could explain why the authors observed such a large spread in the adhesion data. It is to be expected, however, that weaker bonds have larger error due to increased failure modes and also because the measured strengths are within the typical error of bond strength measurements (100s of kPa). Still, a full description of the adhesive testing is required in order for others to reproduce these measurements.

Reviewer #4 (Remarks to the Author):

This study looks at the molecular basis of the glue compounds in the skin secretion of the Madagascan tomato frog, *Dyscophus guineti*, and compares the identified compounds (PRITs and a galectin) with those found in other amphibians species, including a second glue-producing species, *Breviceps mossambicus*. The overall finding is that PRITs are not restricted to glue-producing species with their origin actually predating the frog lineage. The authors therefore rightfully conclude that PRITs would appear to function as an exaptation for the independent evolution of glue within frogs.

Overall, there is little to criticize within the study. All the methods seem to be robust and properly applied and the conclusions are generally solid. Most of my comments revolve around providing more information or justification, especially with respect to some of the evolutionary jargon. My only, major bone of contention are the statements that evolution might have a deterministic side to it, something that I see as being very dangerous.

Otherwise, my comments are minor:

L33: Note that the mussel byssus adhesive is not permanent and the byssus threads can be de-attached from the substrate by the mussel.

L40: "evade" implies that it's complicated or not easy to resolve whereas the first part of the sentence reveals that we really haven't tried.

L42: Although it is widely known, a reference or two is required anyway.

L55: "non-adhesive poison" implies that the glues are poisons, which they really are not. (But note that the use of this phrase in L60 is acceptable because this implication is not made there.) Also, do "most amphibians" produce poisons? I think not.

L56: But isn't this parallel true of the development of poisons in general? There must be other examples as well, all of which makes it that less peculiar.

L71: Add "non-toxis" to again avoid the implication that the glues are poisonous.

L85, 86: Please also provide some measure of the variance here.

L105: Perhaps a bit more clarity here: the serine protease inhibitor was only found in the dorsal skin sample, not the secretions? It's not immediately clear to me.

L107: It should be stated what method was used to identify this protein because the implication was that it was through transcriptomics, which, from the Methods, it does not appear to be.

L113: Please provide some data as the the strength of the hit.

L129: Explain how you design "specific primers" for a gene of unknown sequence? Or were they designed against IgGFcBD? This is also unclear given that only the amino-acid sequence of the latter appears to be known.

L145: Explain more precisely what is meant by disordered.

L184: Include cross-reference to Figure 3C here to identify the species.

L187: The cross-reference to Figure 3A should probably appear here (as well). It would also be helpful if birds could be included here (which are missing in Figure S7). Granted, they could be argued to belong to the same lineage as the rattlesnake, but otherwise all the major lineages work out to merely amphibians, a snake, and a human.

L188: Where are the data to show that the cluster also appears in non-vertebrate deuterostomes?

L193, 194: No idea what either "early diverged" (also L230 to a lesser degree) or especially "highly evolved" mean here. "Early diverged" requires a point of reference (probably within amphibians is meant) and "highly evolved" is simply meaningless. Yes, there are major structural changes in the PRIT-cluster. But what adaptation isn't highly evolved? And given that non-glue producing species also show these structural changes, the more important question becomes what these changes are good for.

L206: Given that *Breviceps* and *Dyscophus* likely diverged about 100 Ma, can simple, further evolution of the repeats be ruled out? What is the evidence for independent origins?

L215: Again, given how important "disorder" seems to be, it is important to define exactly what is meant by this term.

L222: What are the equivalent values in a non-glue producing species? These are needed to put the stated values in context. Maybe other species have similar values.

L246: Although the 2-6x increased expression might be significant, it is still an order of magnitude lower than that found in *Dyscophus*. Any thoughts as to why?

L257: It's probably important to note that only the PRITs show all of these characteristics, meaning that possessing only some of them is sufficient to produce glue.

L257: It would also be helpful to note which of these features distinguish glue-enabling PRITs from "normal" ones. Only (3) has really been discussed up to this point (and also later in the Discussion).

L319: As much as I applaud the evolutionary framework underlying these studies, it must be said that the "evolutionary analyses" are extremely basic in the sense that only two glue-producing species were examined in any detail.

L330: In other words, the glue function is an adaptation of PRITs.

L352: A taxon name is much better than the subjective "advanced frogs".

L358: What does "too coincidental to be true" mean? It almost sounds like an argument for Creationism.

L359: I think it is a huge mistake to refer to a "deterministic" side of evolution, which implies that evolution has directionality. Instead, this is simply a case of parallel evolution of a similar, beneficial adaptation from a common starting point (as argued in L364). Deterministic would imply that these independent adaptations should be much more common and widespread.

L831: The blue bars look more green to me.

Figure 3A: Please (also) provide the scientific names of the organisms so a comparison to Figure 3C can be made.

Figure S7: Note that the sequence LOC120300047 from the rattlesnake clusters within the amphibians sequences. Conspicuous also is the lack of avian sequences.

RESPONSE TO REVIEWERS' COMMENTS

Reviewer #1

*The manuscript by Zaman et al. focuses on the adhesive skin secretions of the Madagascan tomato frog *Dyscophus guineti* (Microhylidae) as a model species. The authors utilized a diverse suite of methods/techniques and experimental approaches to provide direct evidence and fill-in the gaps in the current knowledge in two different directions:*

- i) related to the skin-secreted glue (the anti-predator defence at the level of a frog species) – What is its composition? How the major proteins engage in complex interactions after they are released from the gland? How quickly is the adhesive formed? What are its mechanical properties?*
- ii) related to the origin of the genes encoding skin glue proteins in amphibians (evolution in action involving “recurrent innovation at the molecular scale”) – What is the ancestor that gave rise to the host-defence adhesive components? What are the precise molecular mechanisms that acted on genetic level? Is this an isolated event in *D. guineti*, or it is rather a recurrent innovation that happens independently in different frog lineages?*

In my opinion, this paper is suitable for publication in Nature. The work is technically sound. The methods provide enough details for the work to be reproduced. The authors raise their own specific antibodies that allows them to provide evidence and answer many of the questions related to i) above. The conclusions are supported by the results. Apart from “elucidating the molecular basis of an amphibian adhesive skin secretion”, the authors reveal for the first time “how adhesive secretions arose from a non-adhesive predecessor”. These are original findings and contribute greatly to the current knowledge in the field by filling in important missing gaps and proposing models that can be applied not only to other amphibians but also to other animals that utilize similar modular proteins with multiple domains.

Specific comments and/or questions:

1. In Discussion (p.13, l.314-318) – the authors propose that the trigger for the protein- protein bonding is the change in the physical conditions. Since this is an essential part of the presented paper, could the authors perform an experiment to confirm that?

The specific investigation of protein-protein bonding will have to await the availability of recombinant proteins (something we're currently working on). However, you make a very good point, so we have expanded our pull tests to investigate the role of pressure applied on the glue after its secretion. First, we have added a comparison of tensile strength measurements of glued surfaces joined together at two different pressures (4 kPa and 40 kPa, the latter based on the typical pressure of a snake bite; Figure 1a). This comparison shows a significant difference, indicating that the tensile strength of the glue is indeed pressure-sensitive. Second, we have included scanning electron microscopy (SEM; Figure 1b) and immunolabelling images (Figure 2a) of glue to visualise the effect of applying low vs. moderate pressure. These images show how pressure triggers the increased release of proteinaceous matrix from intactly secreted granules, explaining at least partially how the glue becomes activated upon secretion. We have modified the Results text (lines 95-104 and 153-155) and the relevant section in the Discussion (lines 363-364) accordingly.

2. In Fig. 1b - What is the rationale for illustrating only PRIT-Dg using two different antibodies in glue? Why not show in the main text the presence of galectin-Dg1 in glue and absence from mucus, instead

of having it in Fig. S5? This in fact would be affirmative that the in-house antibodies are recognizing the two important proteins in the glue only.

We agree that adding the galectin images to the main text would make a lot more sense. However, due to the addition of extra panels to Figure 1, we have added the images to Figure 2 instead of Figure 1. This shift makes sense given the structure of the Results section as the immunolabeling is only reported under its second subtitle, along with the other panels in Figure 2. Consequently, the immunolabelled images of both proteins are now in Figure 2a, i.e. in the main manuscript, as you suggested.

3. In Results (p.4, l.34) – the abbreviations GalNAc and GlcNAc are mentioned for the first time. Please spell those out.

Both abbreviations are now spelled out (Results, lines 119-120).

4. In Fig. S6 – Why is the pull-down experiment to analyze the co-precipitation of the two proteins is done only using anti-PRIT-Dg (interdomain repeats)? The anti-PRIT-Dg (IgGFcBD) also should be used in the pull-down step.

Since both sets of antibodies raised against PRIT-Dg target the same protein, with no difference in band size or pattern, we selected the one that consistently produced a stronger signal when detected by Western blots (Figure S11): anti-PRIT-Dg (interdomain repeats). The co-immunoprecipitation experiment was conducted in order to provide a supporting line of evidence for the occurrence of protein-protein interactions in the glue, which we demonstrate using native PAGE (Figure 2d). We believe that our native PAGE results offer a more convincing argument in this regard, as there are fewer experimental variables involved (and are thus more easily reproducible).

5. How would you account for the fact that in your proteomic analysis you were not able to detect the presence of the Kunitz family protease inhibitor as reported by Conlon and Kim (2000) in norepinephrine-stimulated skin secretions? Your only comment was that this inhibitor might be playing a role in the dorsal gland only but not in the secretions.

Reviewers #2 and #4 made related remarks. We removed our original comment since in retrospect, it made little sense because the inhibitor was indeed originally discovered in skin secretion (Conlon & Kim, *Biochem Biophys Res Commun* 2000). Instead, we state that its absence may be due to our choice of proteomic methods (although we consider this unlikely) or due to the previously reported low levels of this protein in the secreted material. The latter explanation is consistent with the study of König *et al.* (2013), who reported that the serine protease inhibitor exhibits “weak intensity in both HPLC and MALDI-TOF”. We have modified the sentence to make this point and cite König *et al.* (2013) (Results, lines 130-133).

General comment:

1. The panels in all figures are labelled with small letters, while in the text they are referred to with capital letters. Please revise for consistency. Additionally, some of the panels are labelled on the top right, while they should all be in the top left of the panel.

Thank you for pointing this out. All references to figure panels in the main text have been changed to lower-case for consistency, and panel labels are now indicated in the top left, as suggested.

2. p.21, l.497 – PRIT should PRIT-Dg.

Indeed, thank you. The suggested correction has been made (line 598).

Reviewer #2

Dear authors, your work is excellent and contributes greatly to the understanding of the mechanisms involved in the adaptation of amphibians, however, some important points need to be reviewed with caution, due to delicate statements that can only be made with these results obtained.

- Between lines 44 and 48 you state that amphibians adapted in two different ways, one group that presents poison, but does not present glue, and a group that presents glue as an adaptation to the absence of poison.

*I consider this statement delicate, given that many anurans, such as *Rhaebo guttatus*, *Rhinella* sp. They have poison on their skin, and they also have a poison that solidifies after the glandular secretion that also looks like glue.*

<https://onlinelibrary.wiley.com/doi/epdf/10.1002/jez.1838>

<https://frontiersinzoology.biomedcentral.com/articles/10.1186/s12983-018-0294-5>

*Another example is found between lines 52 and 54, where you mention hylids as glue producers, and we have important examples of toxicity in casque-headed-frogs, such as *Corythomantis greening* and *Aparasphenodon brunoi*, which belong to the Hylidae family.*

<https://www.cambridge.org/core/journals/journal-of-zoology/article/abs/head-coossification-phragmosis-and-defence-in-the-casqueheaded-tree-frog-corythomantis-greeningi/F8C15562F3F657AAB5639A2D4AA1BE3D>

[https://www.cell.com/fulltext/S0960-9822\(15\)00788-5](https://www.cell.com/fulltext/S0960-9822(15)00788-5)

*It was not our intention to state that **all** hylid species produce glue. We meant some (particularly we were thinking of species of the genus *Trachycephalus*). We modified the relevant sentence in the Introduction to make this clear (lines 54-55).*

Wouldn't there be an intermediate group between the presence of glue and the presence of poison among amphibians during the adaptation process? For example, an amphibian that presents both adaptations. It's possible?

*We admit that our original text was a bit too polarising in making the impression that frog secretion is either poisonous or sticky. As you suggest, we are aware that some skin secretions are actually both. To acknowledge this, we modified all relevant sentences in the Introduction and Results sections that originally could give the impression that poison and glue are mutually exclusive and added a reference (Mailho-Fontana *et al.* *J Exp Zool A Ecol Genet Physiol* 2014) as an example.*

*Has the presence of alkaloids or steroids been studied in *D. guineti*? Has the possibility of this group of animals producing such molecules been ruled out?*

We have not investigated the presence of alkaloids or steroids in the adhesive secretion and thus cannot exclude this possibility. We can, however, confirm that we did not find proteinaceous toxin candidates in either the skin transcriptome or the glue proteome, and have adapted the text to explicitly state this (as well as acknowledge that the presence of alkaloids or steroids is still unknown; lines 133-137).

- Between lines 107 and 108 you state that the PRIT protein has multiple copies of IgGfCBD protein. How can you ensure that even divided by interdomains, they will be translated into a single molecule?

Couldn't they be translated separately? Just like what happens in the venom of snakes like the Jararaca? Which has several coding sequences in tandem, but expresses each protein individually?

<https://www.pnas.org/doi/epdf/10.1073/pnas.2015159118>

To be clear: the statement on lines 107-108 was not based on genomic data (like for the Jararaca), but on a combination of transcriptome and proteome data. As such, we have two lines of evidence confirming the expression of a large protein with multiple IgGFcBD domains. First, the longest transcript, obtained by RNA-seq and subsequent RACE-PCR and primer walking, reveals a ~10000-bp long transcript. Its ORF encodes at least four domains, as illustrated in Figure 2b. Second, our Western blots consistently depict a prominent band far larger than 180 kDa (thus far more than 1600 AA; Figures S6 and S11) - in fact, its size is within the range of 268 kDa and 460 kDa (Figure S2b). This band size cannot be accounted for by the translation of an individual domain, which is less than 400 AA long.

To make sure that readers would not have similar doubts, we now clearly state that the RACE-PCR primers were based on our transcriptome and proteome sequence data.

What is the predicted size of this group of proteins by the transcriptome? Is it similar to the molecular weight obtained in Wersten Blott's analyses?

Since we haven't obtained the full sequence of PRIT-Dg, we can only provide a rough estimate based on transcript data. Based on the available 5'- and 3'-sequences obtained by RACE-PCR and primer walking, we estimate the protein to be approximately 3250 residues long, or roughly 358 kDa (counting 110 Da per residue and not counting glycans). We have added a sentence with this estimate in the Results section (lines 172-174) and added the size estimate on the gel image in Figure S2b. Estimates for other transcript variants (resulting from alternative splicing) are unfortunately not possible.

- Why don't you present the electrophoresis gel of glue and mucus secretion to show the differences in the electrophoretic profile?

We have presented an SDS-PAGE gel comparing mucus and glue in Figure S2, as requested. In addition, we report this in the text (Results, lines 112-113).

Why is the molecular weight standard range not shown in the Wersten Blott image?

The Western blots shown in Figure 2d were carried out under native PAGE conditions, as a result of which protein migration cannot be reliably predicted even with the use of a protein ladder. However, we have indicated molecular weight standards in all supporting gel and blot images.

Did you consider carrying out the electrophoresis under reducing conditions?

We have compared the mucus and glue under both reducing and non-reducing conditions, as you suggested (Figure S2), and report on this in the Results (lines 112-113).

- Regarding proteomics, what was the purpose of performing pre-purification on C-8 before analysis? Do you believe that this type of preparation could have eliminated other important proteins such as the Kunitz-type inhibitor already described for one of the animals? Did you consider a shotgun analysis of the crude material using urea?

The C-8 cartridge is implemented as part of an established in-house protocol for the analysis of highly viscous materials (Janssenswillen *et al.*, 2021) to circumvent concerns about damaging the mass spectrometry equipment (and therefore also accounts for why we did not proceed with a shotgun analysis of the crude material). Moreover, based on our transcriptome data, we expected our primary protein of interest (i.e., the glue protein) to be large, for which the C-8 cartridge is well-suited. Although

we cannot exclude the possibility of the C-8 purification resulting in a loss of other proteins, this seems unlikely to be the case for the serine protease inhibitor, given that the predicted molecular weight of the active peptide is about 6.3 kDa. Proteins of this size should be detectable if present in a sufficiently high amount (see also our responses to remark #5 from Reviewer #1 and remark #8 from Reviewer #4). To acknowledge this, we added a statement in the Results section (lines 130-133).

I strongly recommend accepting this work for publication after minor revisions

What are the noteworthy results?

Yes

Will the work be of significance to the field and related fields?

Yes

How does it compare to the established literature?

To date, no research group has managed to uncover the biochemical composition of glues produced by amphibians with this level of molecular refinement.

Does the work support the conclusions and claims, or is additional evidence needed?

Some revisions cited above

Are there any flaws in the data analysis, interpretation and conclusions? Do these prohibit publication or require revision?

Some revision cited above

Is the methodology sound?

Yes

Does the work meet the expected standards in your field?

Yes

Is there enough detail provided in the methods for the work to be reproduced?

Yes

Reviewer #3

In this work by Zaman et al, the authors use transcriptomic, proteomic, and functional approaches to develop an evolutionary theory for convergent biological glues. Overall, the manuscript is well-written, the work is rigorous, and the ideas are presented clearly. This is an area of wide interest, and relating adhesive biomolecules to their evolution remains a central question in comparing diverse biological strategies. I believe this work is well-suited for this journal and recommend publication pending a few minor points related to the adhesive testing:

- Page 4, line 85: 'tensile strength' is used to describe forces observed from the pull experiments in determining adhesion. Since the authors describe their experiment to be pulling one block on a trolley in shear from one that is fixed, I believe the authors mean shear strength rather than tensile strength. Please correct this terminology.

We actually meant tensile strength, *not* shear strength. Our setup indeed pulled one brick, but it was not in shear from the fixed one. We admit that our original description was too succinct to be clear in this aspect and expanded the Methods section to remedy this (lines 462-465; see also your remark #4 below). In addition, we added a supporting figure (Figure S1), illustrating the setup and showing that we did not measure shear force.

- Page 4, line 85: *The authors refer to tensile strength as ‘determined by both cohesive and adhesive strength’. While this is true, it is an odd thing to point out in the text as every bond strength measurement will rely on both cohesive and adhesive forces. It is also slightly misleading as the authors did not perform separate experiments to determine cohesive versus adhesive strengths, such as determining the bulk modulus of the glue or using more sensitive surface techniques to measure adhesive forces.*

We agree that this statement was somewhat *à propos*, especially without any follow-up experiments (but see our response to your remark #3). It is now removed from the text and instead replaced by an explanation of how we calculated the tensile strength from the pull force (related to your previous remark; lines 465-466 and 469-471).

- *Related to the previous point, and indicated by the authors, understanding the failure mode of the adhesive can shed some light on the mechanisms that are laid out towards the end of the manuscript. This can be inferred from simple images of the samples after fracture, if they are available. If the samples are failing adhesively (glue remains on one side of the bond) then the cohesive forces are stronger than the adhesive forces in the material. On the other hand, if you observe cohesive or mixed mode failure, then the bulk forces are greater than or equal to the adhesive force, respectively. This would be helpful in shaping the hypotheses laid out in the discussion regarding adhesive and cohesive interactions. Generally, adhesive testing should be supported by supplemental images of the samples after fracture.*

Thank you so much for this simple but insightful idea! Since we did not have such photographs, we have repeated the pull tests (see our answer to your remark #4) and taken photographs of the brick surfaces immediately after their separation by the new pull tests. A representative set of these is now added to our manuscript as Figure 1c. They show that after 10 minutes of curing the bricks separate by cohesive failure, but after 60 minutes, the bricks separate due to adhesive failure. This difference is now reported in the Results section (lines 104-111) and we come back to it when hypothesizing on the interactions that support the glue’s adhesion and cohesion (Discussion, lines 322-325).

- *I found that the details in the adhesive testing method section were quite thin. The authors describe testing as one fixed block and another being pulled on a trolley with a strain gauge. Was the trolley pulled at a fixed velocity? Was this by hand or a screw motor? What was the rate? Was pressure applied to the bond during “curing”? Was the volume of adhesive from the frogs fixed? Many of these details could explain why the authors observed such a large spread in the adhesion data. It is to be expected, however, that weaker bonds have larger error due to increased failure modes and also because the measured strengths are within the typical error of bond strength measurements (100s of kPa). Still, a full description of the adhesive testing is required in order for others to reproduce these measurements.*

We admit that the description of our pull tests was rudimentary in the original version of our manuscript. To be honest, we originally conceived these tests as a preliminary step to confirm the adhesive property of *D. guineti* skin secretion compared to a non-adhesive frog and its loss after treatment with proteases. However, your remark made us realise that we could do a much better, more standardised job using a motorised setup. Improving the tests also allowed us to expand the

experiment and test the effect of applying different pressures before curing, as well as take the brick photographs you suggested in your remark #3.

The new results are reported in the Results (lines 85-91 and 153-155) and Figure 1a, while the Methods section has been expanded to include additional details about the experimental design, including pulling velocity, applied pressure before curing and used glue mass (lines 455-462). Glue mass could not be fixed for practical reasons (we had to work fast) but we tried to stay within a range of 6-10 mg and conducted statistical analyses to test whether glue mass variation had an effect on our comparison of different curing times and applied pressures (it didn't). In addition, the pulling setup (including a 3D printed brick holder) is shown in Figure S1.

Reviewer #4

*This study looks at the molecular basis of the glue compounds in the skin secretion of the Madagascan tomato frog, *Dyscophus guineti*, and compares the identified compounds (PRITs and a galectin) with those found in other amphibians species, including a second glue-producing species, *Breviceps mossambicus*. The overall finding is that PRITs are not restricted to glue-producing species with their origin actually predating the frog lineage. The authors therefore rightfully conclude that PRITs would appear to function as an exaptation for the independent evolution of glue within frogs.*

Overall, there is little to criticize within the study. All the methods seem to be robust and properly applied and the conclusions are generally solid. Most of my comments revolve around providing more information or justification, especially with respect to some of the evolutionary jargon. My only, major bone of contention are the statements that evolution might have a deterministic side to it, something that I see as being very dangerous.

Otherwise, my comments are minor:

L33: Note that the mussel byssus adhesive is not permanent and the byssus threads can be de-attached from the substrate by the mussel.

Although this is subject to interpretation (the byssus adhesive is permanent but mussels can detach their byssus through a biointerface with their stem; see Sivasundarampillai *et al.* *Science* 2023), we have removed the word "permanent" from the text to avoid confusion.

L40: "evade" implies that it's complicated or not easy to resolve whereas the first part of the sentence reveals that we really haven't tried.

That's true. The phrase "continues to evade our understanding" has been replaced by the more accurate phrase "remains poorly understood" (lines 41-42).

L42: Although it is widely known, a reference or two is required anyway.

We have added two appropriate references (line 44).

L55: "non-adhesive poison" implies that the glues are poisons, which they really are not. (But note that the use of this phrase in L60 is acceptable because this implication is not made there.) Also, do "most amphibians" produce poisons? I think not.

As a matter of fact (and as pointed out by Reviewer #2) some glues may be poisonous too; toxicity and stickiness are not mutually exclusive. To clarify this, we now clearly state this in the Introduction (lines

51-53). However, in the Results section, we state that we did not find evidence of toxins in *D. guineti* (lines 133-134).

Nevertheless, to avoid a similar interpretation by readers, we have changed the sentence you refer to, from: "As most amphibians (including close relatives of glue-secreting taxa) produce non-adhesive poisons..." to: "As close relatives of these glue-secreting taxa often produce non-adhesive (and often toxic) skin secretions...". The latter also immediately removes our reference to "most amphibians".

L56: But isn't this parallel true of the development of poisons in general? There must be other examples as well, all of which makes it that less peculiar.

Agreed. We have removed the term "peculiar".

L71: Add "non-toxic" to again avoid the implication that the gues are poisonous.

We prefer not to, since it is irrelevant whether the glue protein (not the full secretion) is toxic or not. We're only focusing on the rise of its adhesive property here, not on the loss of toxicity (which we can't prove for the glue in its entirety, as Reviewer #2 correctly points out). Adding this specifically makes for a strange emphasis that may in turn confuse other readers.

L85, 86: Please also provide some measure of the variance here.

After carrying out new tensile strength tests (as outlined above, in response to remarks 3 and 4 of Reviewer #3), we have confirmed that our results are normally distributed and thus provided means and standard deviation values where appropriate (see Figure 1a).

L105: Perhaps a bit more clarity here: the serine protease inhibitor was only found in the dorsal skin sample, not the secretions? It's not immediately clear to me.

Indeed, this is correct: the serine protease inhibitor was only detected in the skin transcriptome, not the glue proteome. Based on your question (and on remark 5 from Reviewer #1 and remark 8 from Reviewer #2), we have modified the text to be less ambiguous and provide two possible explanations for this apparent discrepancy (lines 130-133).

L107: It should be stated what method was used to identify this protein because the implication was that it was through transcriptomics, which, from the Methods, it does not appear to be.

The protein was indeed identified from transcriptomics (RNA-seq, RACE-PCR and primer walking). Proteomic analyses were primarily used to compare its abundance in glue vs. mucus. We have streamlined the Methods section (lines 530-534) to clarify the order of steps taken to identify the two glue proteins.

L113: Please provide some data as the the strength of the hit.

The e-value of the hit (restricted to vertebrates, as specified in the methods section) has been added (Results, line 146).

L129: Explain how you design "specific primers" for a gene of unknown sequence? Or were they designed against IgGFcBD? This is also unclear given that only the amino-acid sequence of the latter appears to be known.

The primers used for RACE-PCR and primer walking were based on the already available transcript sequence data from our RNA-seq library. As we progressively obtained more sequence along the transcript's length, additional primers were designed from newly acquired cDNA sequences and

primer-walked segments. To make this clearer, we have streamlined the relevant sections in Results (lines 163-166) and Methods (lines 550-552).

L145: Explain more precisely what is meant by disordered.

Good point. “Disorder” is a commonly used term in protein biology and indeed has a specific definition. This meaning should be made clear to a broad range of scientists, and so we have added a brief description, along with suitable references, upon its first use (lines 182-183).

L184: Include across-reference to Figure 3C here to identify the species.

We added a reference to Figure 3 (line 224) but prefer not to specify Figure 3c since this would lead to citation of the latter before the first citation of Figure 3a or 3b.

L187: The cross-reference to Figure 3A should probably appear here (as well). It would also be helpful if birds could be included here (which are missing in Figure S7). Granted, they could be argued to belong to the same lineage as the rattlesnake, but otherwise all the major lineages work out to merely amphibians, a snake, and a human.

We have added a bird (the gyrfalcon, *Falco rusticolus*) to the genome comparison (Figure 3a) as well as the phylogenetic tree to show the relationships among the included taxa, and colorised the genes around the IgGFcBD cluster to make the figure clearer. In addition, we have expanded the data sets for the phylogenetic analyses of IgGFcBD and galectin (Figures S9 and S10, respectively) to include this bird as well as a number of other amniotes to the analysis (see below).

L188: Where are the data to show that the cluster also appears in non-vertebrate deuterostomes?

The cluster does *not* appear in non-vertebrate deuterostomes (and we didn’t state so). We only stated that the IgGFcBD domain also exists in proteins of other deuterostomes. This has already been reported and we have added a reference that describes the occurrence of IgGFcBD as a domain with unknown function in non-vertebrate deuterostomes. Since the presence of IgGFcBD in other deuterostomes does not offer any novel insights with respect to glue evolution in amphibians, our phylogenetic analysis is limited to vertebrates. Adding more diverged sequences could lead to spurious results due to mutational saturation.

L193, 194: No idea what either "early diverged" (also L230 to a lesser degree) or especially "highly evolved" mean here. "Early diverged" requires a point of reference (probably within amphibians is meant) and "highly evolved" is simply meaningless. Yes, there are major structural changes in the PRIT-cluster. But what adaptation isn't highly evolved? And given that non-glue producing species also show these structural changes, the more important question becomes what these changes are good for.

Based on your suggestions, we have modified the relevant phrase to: “Phylogenetic analyses of IgGFcBD sequences extracted from these transcriptomes and genomes indicate that PRIT-Dg, along with other amphibian skin-expressed proteins, represents a gene lineage that diverged from other genes in this cluster in an early amphibian ancestor” (lines 231-233). This avoids the (admittedly subjective) terms “early diverged” and “highly evolved”.

L206: Given that Breviceps and Dyscophus likely diverged about 100 Ma, can simple, further evolution of the repeats be ruled out? What is the evidence for independent origins?

We admit that a common origin of the repeats cannot be ruled out, despite their differences in sequence and length. We therefore have adapted the text to reflect either possibility (lines 245-246).

L215: Again, given how important "disorder" seems to be, it is important to define exactly what is meant by this term.

We have addressed this in remark #12; a description has been provided upon its first mentioning.

L222: What are the equivalent values in a non-glue producing species? These are needed to put the stated values in context. Maybe other species have similar values.

We have conducted additional selection analyses in which each of the six non-glue producing species for which we have skin transcriptome libraries is analysed independently by appropriate labelling of foreground branches (equivalent to how we analysed the two glue-secreting species). The results show that the latter two stand out in showing more domain sites under positive selection and have been incorporated in the Results (lines 265-266) and Figure 3c.

L246: Although the 2-6x increased expression might be significant, it is still an order of magnitude lower than that found in *Dyscophus*. Any thoughts as to why?

That is a good question. Although somewhat speculative, we have added a sentence in the Discussion that returns to this difference between both species, postulating that it may be suggestive of either weaker cohesive strength, stronger or more effective binding of galectins, or the potential involvement of other, as-yet unidentified components (lines 325-327).

L257: It's probably important to note that only the PRITs show all of these characteristics, meaning that possessing only some of them is sufficient to produce glue.

We are not sure that we understand this comment correctly. Do you mean that PRITs combine properties of various glue proteins, all of which only have one or few of them? This is not true, so we prefer not to mention this in the text. Similar to PRITs, the majority of known glue proteins have multiple characteristics that make them suitable for adhesion. Some glue proteins from other animals also have additional characteristics that are absent from PRITs (e.g., L-DOPA, modularity with different domains instead of IgGFcBD). In addition, we prefer not to make statements regarding how many of these characteristics would be sufficient to make a glue since adhesion is not an on/off trait. Having only a few of these traits would perhaps result in a weak glue but having all of them could create a stronger, or faster acting glue. Hence, adaptive pressure for a strong or efficient defence glue may explain the presence of multiple characteristics in PRIT.

L257: It would also be helpful to note which of these features distinguish glue-enabling PRITs from "normal" ones. Only (3) has really been discussed up to this point (and also later in the Discussion).

Yes, even though the similarities of glue PRITs with those of other amphibians are discussed later in the Discussion, it makes sense to already allude to it here, directly linked to these traits. Since we prefer not to shuffle the order of the Discussion, we added a sentence mentioning that four of these traits characterise most PRITs and thus they must have evolved *prior* to the origin of glues (closing the sentence with "see further"; lines 305-308). Later we come back to this and, following your remark #24, now mention the term exaptation (lines 379-383).

L319: As much as I applaud the evolutionary framework underlying these studies, it must be said that the "evolutionary analyses" are extremely basic in the sense that only two glue-producing species were examined in any detail.

We sympathise with your opinion but would like to emphasise the multidisciplinary nature of this study. We tried to strike a rational balance between elucidating the structure and function of glue proteins (for which we conducted functional experiments, SEM imaging, transcriptomics and

proteomics and various immuno-assays) and reconstructing their evolution (for which we conducted phylogenetic analyses, genomics, comparative structural analyses and selection analyses).

We have currently worked with two species that are occasionally available through the pet trade. Other known glue-producing species are not, and their addition would require field excursions on different continents, acquisition of the necessary sampling and export permits, additional transcriptome and proteome analyses and redoing the evolutionary analyses. In short, this would be costly and set us back for months, if not year(s). Instead, we believe that comparison of the two current species in a phylogenetic framework with six non-sticky amphibians already provides unprecedented insights into the evolution of amphibian glue, and we feel that our conclusions are justified in light of this dataset. We also hope that our efforts to expand the genomic comparison, improve the phylogenetic analyses and include additional selection analyses will serve to compensate at least partially for the basic impression our study has made.

L330: In other words, the glue function is an exaptation of PRITs.

Indeed. We were doubting whether to introduce this term since it is not generally well known but you convinced us. We have modified the sentence so that the term exaptation is now mentioned (line 382).

L352: A taxon name is much better than the subjective "advanced frogs".

The term "advanced frogs" has been replaced with "neobatrachian frogs" (line 401). To clarify even further what this taxon refers to, we have explicitly added the term to the tree in Figure 3c.

L358: What does "too coincidental to be true" mean? It almost sounds like an argument for Creationism.

We hope it's clear that we did not want to make an argument for creationism. But there's a reason why so many people (scientists and non-scientists alike) are fascinated by patterns such as evolutionary convergence: because at first instance they often seem to defy the odds and challenge us to find an explanation for why they happened anyway. The following sentences in the same paragraph represent our attempt to provide such explanation for the convergent evolution of amphibian glue. What may be a "simple case of parallelism" (see next remark) to evolutionary biologists may not be so for other scientists, including protein biologists, glue specialists, or herpetologists. We have rephrased the sentence to avoid similar allusions to creationism in the future (lines 411-413).

L359: I think it is a huge mistake to refer to a "deterministic" side of evolution, which implies that evolution has directionality. Instead, this is simply a case of parallel evolution of a similar, beneficial adaptation from a common starting point (as argued in L364). Deterministic would imply that these independent adaptations should be much more common and widespread.

We respectfully disagree that determinism implies that evolution has directionality or specifies how often an adaptation should occur. Determinism is a commonly used term in the context of evolutionary convergence (e.g. Losos *et al.* (1998) *Science* **279**(5359): 2115-2118; Blount *et al.* (2018) *Science* **362** (6415): eaam5979; Rincon-Sandoval *et al.* (2020): *PNAS* **117**(52): 33396-33403; Friss *et al.* (2021) *Sci. Rep.* **11**: 11600). Also, our results show that this is not a "simple case of parallel evolution" but that glue evolution in both lineages involved multiple parallel changes in function and gene expression: something that has never been shown before. Granted, given that we have "only" analysed two parallel origins, we have no idea how many more times the same proteins have been recruited in other glue-producing amphibians (although we hint at a third occasion in *Notaden bennetti*). As the term "determinism" seems to stir more controversy than we expected, we prefer to remove it. In addition,

we have rephrased the relevant sentences in the hope that our message is now unambiguously clear (lines 411-413).

L831: The blue bars look more green to me.

Correct, this was a typo. The word “blue” has been changed to “green” in the Figure 3 legend.

Figure 3A: Please (also) provide the scientific names of the organisms so a comparison to Figure 3C can be made.

All common names have been replaced with scientific names in Figure 3.

Figure S7: Note that the sequence LOC120300047 from the rattlesnake clusters within the amphibians sequences. Conspicuous also is the lack of avian sequences.

Yes, sorry about that. In the original figure S7, the labelling of clades was a bit oversimplified and did not take into account the position of sequence *LOC120300047* among amphibian sequences.

In the new figure (now Figure S10), we have modified the clade labelling and used numbers cross-referenced in Figure 3a. In addition, to investigate the position of the rattlesnake sequence among amphibians, we have repeated the phylogenetic analysis on an expanded dataset including more amniotes (e.g., including another squamate: *Zootoca vivipara*) and also conducted ML bootstrapping (with RAxML) besides Bayesian phylogenetic inference. The rattlesnake sequence in question is now found to be closest related to a *Z. vivipara* sequence, and together they are recovered as one of many lineages among which relationships remain unresolved. The latter is not a problem for our results, however, since the PRIT clade - as well as many other gene clades of the cluster (numbered in Figure 3a and S10 - is still supported.

REVIEWERS' COMMENTS

Reviewer #1 (Remarks to the Author):

I have reviewed the revised manuscript by Zaman et al. My comments and suggestions for improvement of the paper have been acknowledged. Additionally, I note the degree of attention devoted to all Reviewers' comments.

In my opinion, the manuscript is even more refined now and provides additional experimental data to support some of the key points. Therefore, I recommend accepting it for publication in Nature.

Just two minor corrections:

p.18, l.408 - I think this is the first time the name of the Australian frog *N. bennetti* is mentioned in the text, so please write it in full.

p.19, l.446 - as you have added a new text "Bouin's solution (Sigma-Aldrich)" here, I suggest you remove the company provider in brackets from l.449.

Reviewer #2 (Remarks to the Author):

The article improved the quality of discussion and the proposed mechanisms are very well discussed.

I only recommend a minor revision:

Two delicate points remain in the text.

In your abstract

"...poisonous skin secretions have been modified into highly adhesive glues..." Reinforces the idea of two distinct groups

Poisonous X Glue

The serine protease inhibitor from *Dyscophus guineti* is a Kunitz-type. This class of inhibitor is present in a wide range of different venoms, classified as toxins

Examples:

YUAN, Chun-Hua, et al. Discovery of a distinct superfamily of Kunitz-type toxin (KTT) from tarantulas. *PLoS one*, v. 3, n. 10, p. e3414, 2008.

ST PIERRE, L. et al. Common evolution of waprins and kunitz-like toxin families in Australian venomous snakes. *Cellular and molecular life sciences*, v. 65, p. 4039-4054, 2008.

SCHWEITZ, Hugues et al. Calcicludine, a venom peptide of the Kunitz-type protease inhibitor family, is a potent blocker of high-threshold Ca²⁺ channels with a high affinity for L-type channels in cerebellar granule neurons. *Proceedings of the National Academy of Sciences*, v. 91, n. 3, p. 878-882, 1994.

ŽUPUNSKI, Vera; KORDIŠ, Dušan. Strong and widespread action of site-specific positive selection in the snake venom Kunitz/BPTI protein family. *Scientific reports*, v. 6, n. 1, p. 37054, 2016.

BAYRHUBER, Monika et al. Conkunitzin-S1 is the first member of a new Kunitz-type neurotoxin family: structural and functional characterization. *Journal of Biological Chemistry*, v. 280, n. 25, p. 23766-23770, 2005.

CHEN, Zong-Yun et al. Hg1, novel peptide inhibitor specific for Kv1.3 channels from first scorpion

Kunitz-type potassium channel toxin family. *Journal of biological chemistry*, v. 287, n. 17, p. 13813-13821, 2012.

Why do you not indicate that this inhibitor present in the studied amphibian is a toxin?

Considering the neuromodulatory activity of some representatives of this class, could this inhibitor have a direct effect on the predator?

Some examples:

GUO, Xiaopu et al. A novel Kunitz-type neurotoxin peptide identified from skin secretions of the frog *Amolops loloensis*. *Biochemical and Biophysical Research Communications*, v. 528, n. 1, p. 99-104, 2020.

Reviewer #3 (Remarks to the Author):

The authors have addressed my concerns in this revision, which mainly centered around the treatment of adhesive testing. They have improved upon their testing method, clearly describe and document the testing setup in the SI, and use the new data to support their discussion on adhesive and cohesive interactions. Overall, the tensile tests now support some of the interaction regimes that the authors lay out at the end. They observe that cohesive forces in the material strengthen over time upon pressure activation, causing the bond to switch from cohesive failure to adhesive failure with longer cure times.

This is all supported by the fact that without pressure activation, the adhesive fails to bond to the material surface at all. The data tells a compelling story about how pressure activates components in the adhesive that form cross-links over a longer timeframe. Given the extensive new adhesive testing, I recommend this work for publication.

Reviewer #4 (Remarks to the Author):

The authors have done an excellent job addressing my concerns from the previous round of reviews and the additional information and analyses that they provide enhance the MS greatly.

I really have only some (really) minor comments:

L45: Use "clade" instead of the taxonomic "class"?

L102: I really like this new addition, but am slightly confused as to what the "glue" is and where it's coming from especially with respect to the granules. The start of the paragraph makes it sound like the glue contains granules whereas L102 makes it sound like the glue is contained within the granules, which then release more of it under more pressure.

L278: Remove the word "a" before "orthologous".

L313: Although the topic of the granules is raised again on PG 16, I think some discussion of it belongs here as well. Again, it is unclear to me what the difference between the glue and the granules are and what the granules contain.

L323: Should "cohesive" be "cohesion" or "cohesive strength"?

L379: Insert "it" before "shared".

RESPONSE TO REVIEWERS' COMMENTS

Reviewer #1

I have reviewed the revised manuscript by Zaman et al. My comments and suggestions for improvement of the paper have been acknowledged. Additionally, I note the degree of attention devoted to all Reviewers' comments.

In my opinion, the manuscript is even more refined now and provides additional experimental data to support some of the key points. Therefore, I recommend accepting it for publication in Nature. Just two minor corrections:

1) p.18, l.408 - *I think this is the first time the name of the Australian frog *N. bennetti* is mentioned in the text, so please write it in full.*

*We provide the full name of *Notaden bennetti* during its first appearance in the text, which is in the introduction (line 60). The second occurrence is mentioned much later, in the discussion, and as such we have not expanded the species name in this latter instance.*

2) p.19, l.446 - *as you have added a new text "Bouin's solution (Sigma-Aldrich)" here, I suggest you remove the company provider in brackets from l.449.*

Thank you for pointing this out. The company provider has been removed in the second case (line 445).

Reviewer #2

The article improved the quality of discussion and the proposed mechanisms are very well discussed.

I only recommend a minor revision:

Two delicate points remain in the text.

In your abstract

"...poisonous skin secretions have been modified into highly adhesive glues..." Reinforces the idea of two distinct groups

Poisonous X Glue

Indeed, we have reformulated the above sentence to omit the mention of poison entirely (lines 19-20). We believe the abstract in its newly revised form better reflects the background of our study.

*The serine protease inhibitor from *Dyscophus guineti* is a Kunitz-type. This class of inhibitor is present in a wide range of different venoms, classified as toxins*

Examples:

*YUAN, Chun-Hua, et al. Discovery of a distinct superfamily of Kunitz-type toxin (KTT) from tarantulas. *PloS one*, v. 3, n. 10, p. e3414, 2008.*

*ST PIERRE, L. et al. Common evolution of waprin and kunitz-like toxin families in Australian venomous snakes. *Cellular and molecular life sciences*, v. 65, p. 4039-4054, 2008.*

SCHWEITZ, Hugues et al. Calcicludine, a venom peptide of the Kunitz-type protease inhibitor family, is a potent blocker of high-threshold Ca²⁺ channels with a high affinity for L-type channels in cerebellar granule neurons. *Proceedings of the National Academy of Sciences*, v. 91, n. 3, p. 878-882, 1994.

ŽUPUNSKI, Vera; KORDIŠ, Dušan. Strong and widespread action of site-specific positive selection in the snake venom Kunitz/BPTI protein family. *Scientific reports*, v. 6, n. 1, p. 37054, 2016.

BAYRHUBER, Monika et al. Conkunitzin-S1 is the first member of a new Kunitz-type neurotoxin family: structural and functional characterization. *Journal of Biological Chemistry*, v. 280, n. 25, p. 23766-23770, 2005.

CHEN, Zong-Yun et al. Hg1, novel peptide inhibitor specific for Kv1.3 channels from first scorpion Kunitz-type potassium channel toxin family. *Journal of biological chemistry*, v. 287, n. 17, p. 13813-13821, 2012.

1) Why do you not indicate that this inhibitor present in the studied amphibian is a toxin?

We prefer to avoid any such statements as the *D. guineti* Kunitz protein has never been proven to function as a toxin. For example, König et al. (2013) demonstrated that the inhibitor has no antimicrobial (= cytotoxic) activity; instead, the authors propose that the skin secretion produced by this species serves a role in mechanical defence, rather than as a toxin.

KÖNIG, Enrico et al. Molecular cloning of the trypsin inhibitor from the skin secretion of the Madagascan Tomato Frog, *Dyscophus guineti* (Microhylidae), and insights into its potential defensive role. *Organisms Diversity & Evolution*, v. 13, n. 3, p. 453-461, 2013.

2) Considering the neuromodulatory activity of some representatives of this class, could this inhibitor have a direct effect on the predator?

Some examples:

GUO, Xiaopu et al. A novel Kunitz-type neurotoxin peptide identified from skin secretions of the frog *Amolops loloensis*. *Biochemical and Biophysical Research Communications*, v. 528, n. 1, p. 99-104, 2020.

Although this is possible, a neurotoxic function has never been demonstrated for this protein. While Kunitz-type proteins in some venomous taxa exhibit neurotoxic activity, in many others this is not the case. In humans, for instance, over 20 Kunitz domain-containing proteins exist – none of which are toxins (e.g., SPINT1-4, papilin, TFPI1-2).

Reviewer #3

The authors have addressed my concerns in this revision, which mainly centered around the treatment of adhesive testing. They have improved upon their testing method, clearly describe and document the testing setup in the SI, and use the new data to support their discussion on adhesive and cohesive interactions. Overall, the tensile tests now support some of the interaction regimes that the authors lay out at the end. They observe that cohesive forces in the material strengthen over time upon pressure activation, causing the bond to switch from cohesive failure to adhesive failure with longer cure times.

This is all supported by the fact that without pressure activation, the adhesive fails to bond to the material surface at all. The data tells a compelling story about how pressure activates components in

the adhesive that form cross-links over a longer timeframe. Given the extensive new adhesive testing, I recommend this work for publication.

Thank you for the insightful yet pragmatic comments you have kindly provided. We believe our manuscript has improved greatly thanks to your suggestions, which enabled us to expand the study's adhesive testing dimension and, ultimately, provide more convincing evidence to support our conclusions.

Reviewer #4

The authors have done an excellent job addressing my concerns from the previous round of reviews and the additional information and analyses that they provide enhance the MS greatly.

I really have only some (really) minor comments:

1) L45: Use "clade" instead of the taxonomic "class"?

We agree that the term is better suited for an evolutionary study, and have made the substitution accordingly (line 43).

2) L102: *I really like this new addition, but am slightly confused as to what the "glue" is and where it's coming from especially with respect to the granules. The start of the paragraph makes it sound like the glue contains granules whereas L102 makes it sound like the glue is contained within the granules, which then release more it it under more pressure.*

We acknowledge that the phrasing may perhaps be susceptible to misinterpretation; as such, we have reworded the relevant sentences to improve clarity (lines 100 to 102).

3) L278: Remove the word "a" before "orthologous".

We have corrected this typo (line 276).

4) L313: *Although the topic of the granules is raised again on PG 16, I think some discussion of it belongs here as well. Again, it is unclear to me what the difference between the glue and the granules are and what the granules contain.*

To improve consistency and coherence, we have adapted this section in a way that circles back to the changes made in response to remark #2 (lines 309 to 312).

5) L323: Should "cohesive" be "cohesion" or "cohesive strength"?

Indeed. We have corrected the term to "cohesion" (line 320).

6) L379: Insert "it" before "shared".

We believe the original sentence sounds more harmonious with respect to the overall text (line 376). However, we ultimately leave this to the editor's discretion.